# Serial Decoders-Based Auto-Encoders for Image Reconstruction

**Honggui Li [1,*] , Maria Trocan [2], Mohamad Sawan [3,4] and Dimitri Galayko [5]**

1   School of Information Engineering, Yangzhou University, Yangzhou 225000, China
2   Institut Supérieur d'Électronique de Paris, 92130 Issy Les Moulineaux, France
3   Department of Electrical Engineering, Polystim Neurotechnology Laboratory, Polytechnique Montreal, Montreal, QC H3T 1J4, Canada
4   CenBRAIN Lab, School of Engineering, Westlake University, Hangzhou 310024, China
5   Laboratoire d'Informatique de Paris 6, Sorbonne University, 75005 Paris, France
*   Correspondence: hgli@yzu.edu.cn; Tel.: +86-189-527-81178

**Featured Application: The proposed method can be utilized for highly efficient data compression, signal-compressed sensing, data restoration, etc.**

**Abstract:** Auto-encoders are composed of coding and decoding units; hence, they hold an inherent potential of being used for high-performance data compression and signal-compressed sensing. The main disadvantages of current auto-encoders comprise the following aspects: the research objective is not to achieve lossless data reconstruction but efficient feature representation; the evaluation of data recovery performance is neglected; it is difficult to achieve lossless data reconstruction using pure auto-encoders, even with pure deep learning. This paper aims at performing image reconstruction using auto-encoders, employs cascade decoders-based auto-encoders, perfects the performance of image reconstruction, approaches gradually lossless image recovery, and provides a solid theoretical and applicational basis for auto-encoders-based image compression and compressed sensing. The proposed serial decoders-based auto-encoders include the architectures of multi-level decoders and their related progressive optimization sub-problems. The cascade decoders consist of general decoders, residual decoders, adversarial decoders, and their combinations. The effectiveness of residual cascade decoders for image reconstruction is proven in mathematics. Progressive training can efficiently enhance the quality, stability, and variation of image reconstruction. It has been shown by the experimental results that the proposed auto-encoders outperform classical auto-encoders in the performance of image reconstruction.

**Keywords:** auto-encoders; serial decoders; cascade decoders; general decoders; residual decoders; adversarial decoders; image reconstruction

---

## 1. Introduction

Since deep learning achieves the rules and features from input data using multi-layer stacked neural networks in a highly efficient manner, it has garnered unprecedented successful research and applications in the domains of data classification, recognition, compression, and processing [1,2]. Although the theoretical research and engineering applications of deep learning have matured, there is still much room to improve, and deep learning has not yet attained the requirements for general artificial intelligence [1,2]. Hence, it is incumbent on researchers to utilize deep learning to upgrade the performance of data compression and signal-compressed sensing.

Data reconstruction is the foundation of data compression and signal-compressed sensing. It contains multifarious meanings in understanding from a broad sense. In this paper, data reconstruction denotes high-dimensional original data being initially mapped into a low-dimensional space and then being recovered. Although the classical methods of data compression and signal-compressed sensing are full-blown, it is still necessary

to investigate new algorithms that are based on deep learning. Currently, merely some of the components of traditional data compression methods such as prediction coding, transformation coding, and quantization coding are being replaced by deep learning methods. The principal difficulty is that lossless data reconstruction of pure deep learning-based methods has not yet been attained. In consideration of the powerful capabilities of deep learning, this article will explore new approaches to data reconstruction via pure deep-learning based methods.

Auto-encoders (AE) are a classical architecture of deep neural networks, which initially project high-dimensional data into a low-dimensional latent space according to a given rule, and then reconstruct the original data from latent space while minimizing reconstruction error [3–6]. Auto-encoders possess many theoretical models, including the following: sparse auto-encoders, convolutional auto-encoders, variational auto-encoders (VAE), adversarial auto-encoders (AAE), Wasserstein auto-encoders (WAE), graphical auto-encoders, extreme learning auto-encoders, integral learning auto-encoders, inverse function auto-encoders, recursive or recurrent auto-encoders, double or couple auto-encoders, de-noising auto-encoders, generative auto-encoders, fuzzy auto-encoders, non-negative auto-encoders, binary auto-encoders, quantum auto-encoders, linear auto-encoders, blind auto-encoders, group auto-encoders, kernel auto-encoders, etc. [3–6]. Some of the theoretical frameworks of traditional auto-encoders are collected in Table 1. Auto-encoders have garnered extensive research and applications in the domains of classification, recognition, encoding, sensing, and processing [3–6]. Since auto-encoders comprise encoding and decoding units, they hold the potential of being applied to high-performance data compression and signal-compressed sensing [7,8]. Classical auto-encoders shall be referred to as narrow auto-encoders. Other deep learning-based methods of data compression and signal-compressed sensing shall be referred to as generalized auto-encoders, because they contain encoding and decoding components, and each component can introduce an auto-encoder unit [9,10]. Narrow and generalized auto-encoders-based approaches of data compression and signal-compressed sensing can provide better performance in data reconstruction than the classical approaches [7–10].

**Table 1.** Some of the theoretical frameworks of traditional auto-encoders.

| Auto-Encoders | References |
|---|---|
| Variational auto-encoders | [4] |
| Adversarial auto-encoders | [11] |
| Convolutional auto-encoders | [12] |
| Quantum auto-encoders | [13] |
| Sparse auto-encoders | [14] |
| Wasserstein auto-encoders | [15] |
| Graphical auto-encoders | [16] |

However, current research in auto-encoders exhibits the following problems: the research objective is not to achieve lossless data reconstruction but efficient feature representation; independent evaluation of the performance of data reconstruction is neglected; the performance of data reconstruction needs to be improved; it is difficult to attain lossless data reconstruction [17,18]. For instance, the performance of data reconstruction using AAE, one of the most advanced auto-encoders, is shown in Figure 1 [11]. The horizontal axis is the dimension of the latent space and the vertical axis is the average structural similarity (SSIM) of reconstructed images in comparison with original images. It is indicated in Figure 1 that the performance in data reconstruction of AAE increases while the dimension of hidden space increases, making it difficult to achieve lossless data reconstruction. Currently, pure deep learning-based methods of data compression and signal-compressed sensing cannot attain lossless data reconstruction.

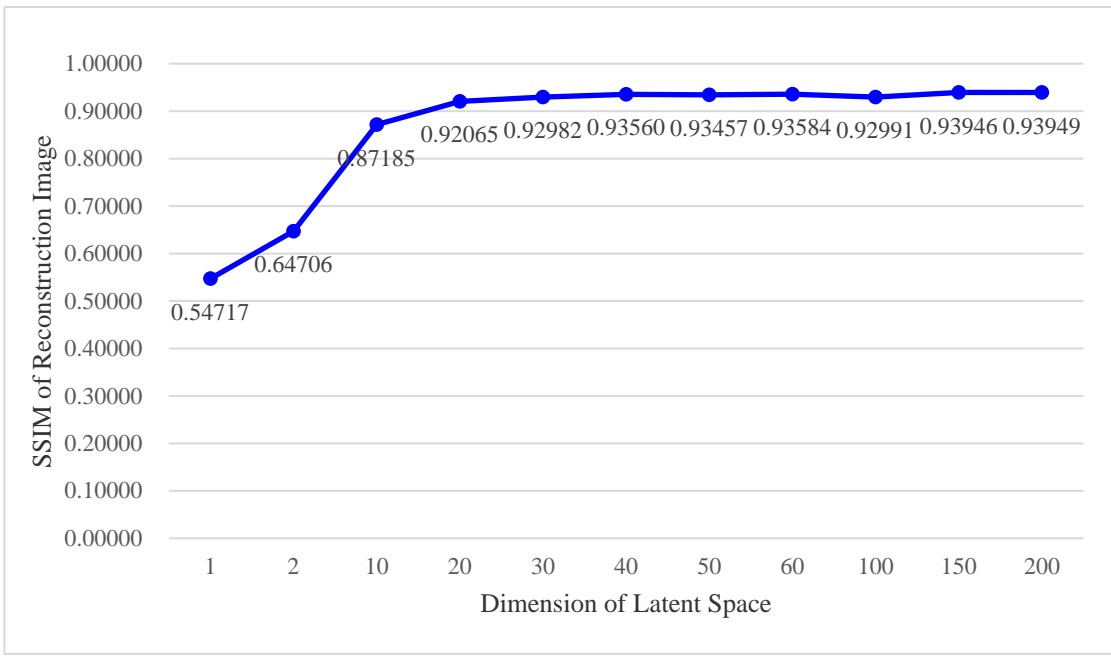

**Figure 1.** Data reconstruction performance using AAE.

This manuscript attempts to regard lossless data reconstruction as a research goal of auto-encoders, and independently assesses the performance of data reconstruction of auto-encoders, enhances the quality of data reconstruction of auto-encoders, gradually approaches lossless data reconstruction of auto-encoders, and builds a solid theoretical and applicational foundation of data compression and signal-compressed sensing for auto-encoders.

This article proposes serial decoders-based auto-encoders for image reconstruction. The main contribution of this paper is to introduce cascade decoders into auto-encoders, including theoretical architectures and optimization problems. The optimization problems are divided into sequential sub-problems in order to progressively train the deep neural networks. Progressive training can efficiently improve the quality, stability, and variation of image reconstruction. The components of serial decoders consist of general decoders, residual decoders, adversarial decoders, and their combinations. The effectiveness of residual serial decoders for image reconstruction is proven in mathematics. Since AAE, VAE, and WAE are state-of-the-art auto-encoders, this article focuses on their cascade decoders-based versions.

The rest of this article is organized as follows: the related research is summarized in Section 2, theoretical foundations are established in Section 3, simulation experiments are designed in Section 4, and final conclusions are drawn in Section 5.

## 2. Related Research

Narrow auto-encoders-based data compression and signal compressing have progressed rapidly [7,8,12–14,19–21]. Firstly, auto-encoders have been studied and applied in the compression of medical signals, navigation data, and quantum states [7,12,13,19]. For example, Wu Tong et al. proposed an auto-encoders-based compression method of brain neural signals [7]. Yildirim Ozal et al. utilized convolutional auto-encoders to compress electrocardio signals [12]. Lokukaluge P. Perera et al. employed linear auto-encoders to compress navigation data [19]. Romero Jonathan et al. used quantum auto-encoders to compress quantum states [13]. Secondly, auto-encoders have already been studied and applied in the compressed sensing of biomedical signals, images, and sensor data [8,14,20,21]. For instance, Gogna Anupriya et al. utilized stacked and label-consistent auto-encoders to reconstruct electrocardio signals and electroencephalograms [8]. Biao Sun et al. used binary

auto-encoders for compressed sensing of neural signals [20]. Majumdar Angshul utilized auto-encoders to reconstruct magnetic resonant images [21]. Han Tao et al. adopted sparse auto-encoders to reconstruct sensor signals [14].

Important developments in narrow auto-encoders also include the following: the Wasserstein auto-encoder, the inverse function auto-encoder, and graphical auto-encoders [15,16,22]. For example, Ilya Tolstikhin et al. raised Wasserstein auto-encoders, which are generalized adversarial auto-encoders, and utilized the Wasserstein distance to measure the difference between data model distribution and target distribution, in order to gain better performance in data reconstruction than classical variational auto-encoders and adversarial auto-encoders [15]. Yimin Yang et al. employed inverse activation function and pseudo inverse matrix to achieve the analysis representation of the network parameters of auto-encoders for dimensional reduction and reconstruction of image data, and hence improve the data reconstruction performance of auto-encoders [22]. Majumdar Angshul presented graphical auto-encoders, used graphical regularization for data de-noising, clustering and classification, and consequently attained better data reconstruction performance than classical auto-encoders [16].

Generalized auto-encoders-based data compression and signal-compressed sensing have also achieved significant evolution [9,10,23]. These methods usually utilize multi-level auto-encoders to overcome the disadvantage that single-level auto-encoders have in being unable to achieve lossless data reconstruction; these methods use auto-encoders to replace one unit of the classical data compression and signal-compressed sensing model, such as the prediction, transformation, or quantization unit of data compression, as well as the measurement or recovery unit of signal-compressed sensing. For instance, George Toderici et al. applied two-level auto-encoders for image compression. The first-level auto-encoders compress image blocks, and the second-level auto-encoders compress the recovery residuals of the first-level auto-encoders. This approach makes up for the disadvantage that single-level auto-encoders have in being unable to implement lossless data reconstruction to a great degree [9]. Oren Rippel et al. adopted multi-level auto-encoders to implement the transformation coding unit of video compression. The first-level auto-encoders compress the prediction residuals, and the next-level auto-encoders compress the reconstruction residuals of the previous-level auto-encoders to a great extent [10]. Majid Sepahvand et al. employed auto-encoders to implement the prediction coding unit of compressed sensing of sensor signals [23].

The main research advances in generalized auto-encoders also comprise the use of other architectures of deep neural networks to implement data compression and signal-compressed sensing [24–27]. In data compression, these methods usually wield deep neural networks to substitute the prediction, transformation, or quantization units of classical methods. In signal-compressed sensing, these methods usually implement deep neural networks to substitute the measurement or recovery units of classical methods. For example, Jiahao Li et al. utilized fully-connected deep neural networks to realize the intra prediction coding unit of video compression [25]. Guo Lu et al. adopted deep convolutional neural networks to replace the transformation coding unit of video compression [26]. Wenxue Cui et al. employed a deep convolutional neural network to accomplish the sampling and reconstruction units of image-compressed sensing [27].

This paper focuses on narrow auto-encoders, incorporates multi-level decoders into auto-encoders, and boosts the performance of data reconstruction. To the best of our knowledge, cascade decoders in auto-encoders have never been studied. Although serial auto-encoders have already been investigated, serial decoders in auto-encoders play a more important role in data reconstruction. In addition, Tero Karras et al. progressively trained generative adversarial networks by gradually increasing the layer numbers of generator and discriminator in order to improve the quality, stability, and variability in data reconstruction [28]. This method will be borrowed for progressively training the proposed cascade decoders-based auto-encoders. The proposed training method gradually increases the decoders of auto-encoders. It is difficult for us to train stable auto-encoders

using multiple decoders and large hypo-parameters. However, it is easier for us to train a stable unit of auto-encoders using a single decoder and small hypo-parameters. A decoder can merely learn low image variation, but serial decoders can learn high image variation. Hence, progressive training can efficiently strengthen the quality, stability, and variability of image reconstruction.

## 3. Theory

### 3.1. Notations and Abbreviations

For the convenience of content description, parts of the mathematical notations and abbreviations adopted in this manuscript are listed in Table 2.

**Table 2.** Mathematical notations and abbreviations.

| Notations and Abbreviations | Meanings |
|---|---|
| AE | auto-encoders |
| AAE/VAE/WAE | adversarial/variational/Wasserstein AE |
| CD | cascade decoders |
| GCD/RCD/ACD/RACD | general/residual/adversarial/residual-adversarial CD |
| CD/GCDAE/RCDAE/ACDAE/RACDAE | CD/GCD/RCD/ACD/RACD-based AE |
| CDAAE/GCDAAE/RCDAAE/ACDAAE/RACDAAE | CD/GCD/RCD/ACD/RACD-based AAE |
| CDVAE/GCDVAE/RCDVAE/ACDVAE/RACDVAE | CD/GCD/RCD/ACD/RACD-based VAE |
| CDWAE/GCDWAE/RCDWAE/ACDWAE/RACDWAE | CD/GCD/RCD/ACD/RACD-based WAE |
| E/D/DC | encoder/decoder/discriminator |
| **x**/**y**/**z** | original/reconstructed/latent sample |

### 3.2. Recall of Classical Auto-Encoders

The architecture of classical auto-encoders is illustrated in Figure 2. Classical auto-encoders are composed of two units: encoder and decoder. The encoder reduces the high-dimensional input data to a low-dimensional representation, and the decoder reconstructs the high-dimensional data from the low-dimensional representation. The classical auto-encoder can be described by the following formulas:

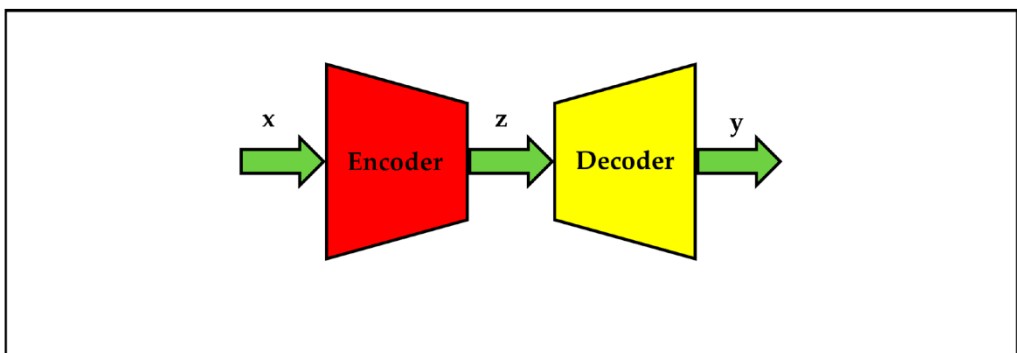

**Figure 2.** The architecture of classical auto-encoders.

$$\begin{aligned} \mathbf{z} &= \mathrm{E}(\mathbf{x}) \\ \mathbf{y} &= \mathrm{D}(\mathbf{z}) \\ \mathbf{x}, \mathbf{y} &\in \mathrm{R}^{\mathrm{H}}; \mathbf{z} \in \mathrm{R}^{\mathrm{L}} \\ \mathrm{H} &\gg \mathrm{L} \end{aligned}$$

$$(1)$$

where the terms are defined as follows:

**x** is the high-dimensional input data. Taking image data as an example, **x** is the normalized version of original image for the convenience of numerical computation; each element of the original image is an integer in the range [0, 255]; each element of **x** is a real number in the range [0, 1] or [−1, +1]; **x** with elements in the range [0, 1] can be understood as probability variables; **x** can also be regarded as a vector which is a reshaping version of an image matrix.

$\mathbf{z}$ is the low-dimensional representation in a latent space.
$\mathbf{y}$ is the high-dimensional data, such as a reconstruction image.
E is the encoder.
D is the decoder.
H is the dimension of $\mathbf{x}$ or $\mathbf{y}$; for image data, H is equal to the product of image width and height.
L is the dimension of $\mathbf{z}$ and L is far less than H.

The classical auto-encoders can be resolved by the following optimization problem:

$$(\theta, \mathbf{z}, \mathbf{y}) = \underset{\theta, \mathbf{y}, \mathbf{z}}{\operatorname{argmin}} \|\mathbf{y} - \mathbf{x}\|_2^2$$
$$\text{s.t. } \mathbf{z} = E(\mathbf{x}), \mathbf{y} = D(\mathbf{z}), C_z = \|\mathbf{z} - \mathbf{z}_g\|_2^2 < \delta_z, C_y = \|\mathbf{y} - \mathbf{D}_y \mathbf{s}_y\|_2^2 + \lambda_y \|\mathbf{s}_y\|_1 < \delta_y \tag{2}$$

where the terms are defined as follows:

$\theta$ are the parameters of auto-encoders, including the parameters of the encoder and decoder.
$C_z$ is the constraint on low-dimensional representation $\mathbf{z}$; for example, $\mathbf{z}$ satisfies a given probability distribution; it has been considered to match a known distribution by classical adversarial auto-encoders, variational auto-encoders, and Wasserstein auto-encoders.
$\mathbf{z}_g$ is a related variable which meets a given distribution.
$\delta_z$ is a small constant.
$C_y$ is the constraint on high-dimensional reconstruction data $\mathbf{y}$; for instance, $\mathbf{y}$ meets a prior of local smoothness or non-local similarity. Auto-encoders require y to reconstruct $\mathbf{x}$ based on the prior to a great extent; other constraints, such as sparsity and low-rank properties of high-dimensional reconstruction data can also be utilized; hereby, sparse prior is taken as an example.
$\mathbf{D}_y$ is a matrix of sparse dictionary.
$\mathbf{s}_y$ is a vector of sparse coefficients.
$\lambda_y$ is a small constant.
$\delta_y$ is a small constant.

### 3.3. Proposed Cascade of Decoders-Based Auto-Encoders

The framework of the proposed cascade decoders-based auto-encoders (CDAE) is exhibited in Figure 3. The framework consists of two components: encoder and cascade decoders. The encoder is similar to that in the classical auto-encoder. Cascade decoders comprise N serial decoders, from decoder 1 to N. The framework can be depicted by the following expressions:

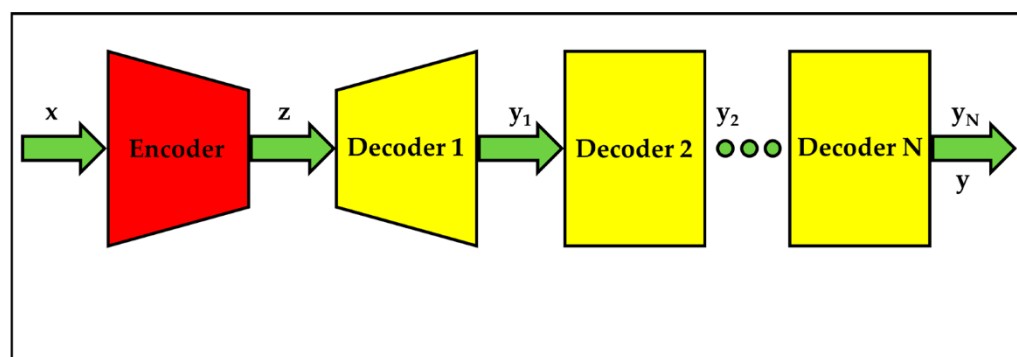

**Figure 3.** The architecture of cascade decoders-based auto-encoders.

$$\begin{aligned} \mathbf{z} &= E(\mathbf{x}) \\ \mathbf{y}_n &= \begin{cases} D_1(\mathbf{z}), n = 1 \\ D_n(\mathbf{y}_{n-1}), n = 2, \dots, N \end{cases}, \mathbf{y} = \mathbf{y}_N \end{aligned} \tag{3}$$

where the terms are defined as follows:

$D_n$ is the nth decoder;

$\mathbf{y}_n$ is the reconstruction data of $D_n$.

The reconstruction data can be solved by the following optimization problem:

$$
\begin{aligned}
(\theta; \mathbf{z}; \mathbf{y}_1, \cdots, \mathbf{y}_N; \mathbf{y}) &= \underset{\theta; \mathbf{z}; \mathbf{y}_1, \cdots, \mathbf{y}_N; \mathbf{y}}{\mathrm{argmin}} \sum_{n=1}^{N} \|\mathbf{y}_n - \mathbf{x}\|_2^2 \\
\text{s.t. } \mathbf{z} &= E(\mathbf{x}); \mathbf{y}_1 = D_1(\mathbf{z}); \mathbf{y}_n = D_n(\mathbf{y}_{n-1}), n = 2, \ldots, N; \\
&\quad C_{\mathbf{z}}; C_{\mathbf{y}_n}, n = 1, \ldots, N; \mathbf{y} = \mathbf{y}_N
\end{aligned}
\tag{4}
$$

where the terms are defined as follows:

$\theta$ are the parameters of cascade decoders-based auto-encoders;
$C_{yn}$ is the constraint on $\mathbf{y}_n$.

For the purpose of gradually and serially training cascade decoders-based auto-encoders, the optimization problem in Equation (4) can be divided into the following suboptimization problems:

$$
\begin{aligned}
(\theta_1; \mathbf{z}; \mathbf{y}_1) &= \underset{\theta_1; \mathbf{z}; \mathbf{y}_1}{\mathrm{argmin}} \|\mathbf{y}_1 - \mathbf{x}\|_2^2 \\
(\theta_2; \mathbf{y}_2) &= \underset{\theta_2; \mathbf{y}_2}{\mathrm{argmin}} \|\mathbf{y}_2 - \mathbf{x}\|_2^2 \\
&\cdots\cdots \\
(\theta_N; \mathbf{y}_N; \mathbf{y}) &= \underset{\theta_N; \mathbf{y}_N; \mathbf{y}}{\mathrm{argmin}} \|\mathbf{y}_N - \mathbf{x}\|_2^2 \\
\text{s.t. } \mathbf{y} &= E(\mathbf{x}); \mathbf{y}_1 = D_1(\mathbf{z}); \mathbf{y}_n = D_n(\mathbf{y}_{n-1}), n = 2, \ldots, N; \\
&\quad C_{\mathbf{z}}; C_{\mathbf{y}_n}, n = 1, \ldots, N; \mathbf{y} = \mathbf{y}_N
\end{aligned}
\tag{5}
$$

where the terms are defined as follows:

$\theta_1$ are the parameters of encoder and decoder 1;
$\theta_2, \ldots,$ and $\theta_N$ are the parameters of decoder 2 to N.

The proposed cascade decoders include general cascade decoders, residual cascade decoders, adversarial cascade decoders and their combinations. The general cascade decoders-based auto-encoders (GCDAE) have already been introduced in Figure 3. The other cascade decoders are elaborated in the following sections.

### 3.3.1. Residual Cascade Decoders-Based Auto-Encoders

The infrastructure of residual cascade decoders-based auto-encoders (RCDAE) is demonstrated in Figure 4. The blue signal flow is for the training phase, and the green signal flow is for both phases of training and testing. Each decoder is a residual module. This architecture is different from the traditional residual network (ResNet) because the former has an extra training channel for residual computation.

The reconstruction data can be resolved by the following optimization problem:

$$
\begin{aligned}
(\theta; \mathbf{z}; \mathbf{r}_1, \cdots, \mathbf{r}_N; \mathbf{y}_1, \cdots, \mathbf{y}_N; \mathbf{y}) &= \underset{\theta; \mathbf{z}; \mathbf{y}_1, \cdots, \mathbf{y}_N; \mathbf{y}}{\mathrm{argmin}} \sum_{n=1}^{N} \|\mathbf{x} - \mathbf{y}_{n-1} - \mathbf{r}_n\|_2^2 \\
\text{s.t. } \mathbf{z} &= E(\mathbf{x}); \mathbf{r}_1 = D_1(\mathbf{z}); \mathbf{r}_n = D_n(\mathbf{y}_{n-1}), n = 2, \ldots, N; \\
C_{\mathbf{z}}; C_{\mathbf{y}_n}, n &= 1, \ldots, N; \mathbf{y}_0 = 0; \mathbf{y}_n = \mathbf{r}_n + \mathbf{y}_{n-1}, n = 1, \ldots, N; \mathbf{y} = \mathbf{y}_N
\end{aligned}
\tag{6}
$$

where the variables are defined as follows:

$\mathbf{r}_n$ is the residual sample between $\mathbf{x}$ and $\mathbf{y}_n$;
$\mathbf{y}_0$ is the zero sample;
$\mathbf{y}$ is the final reconstruction sample.

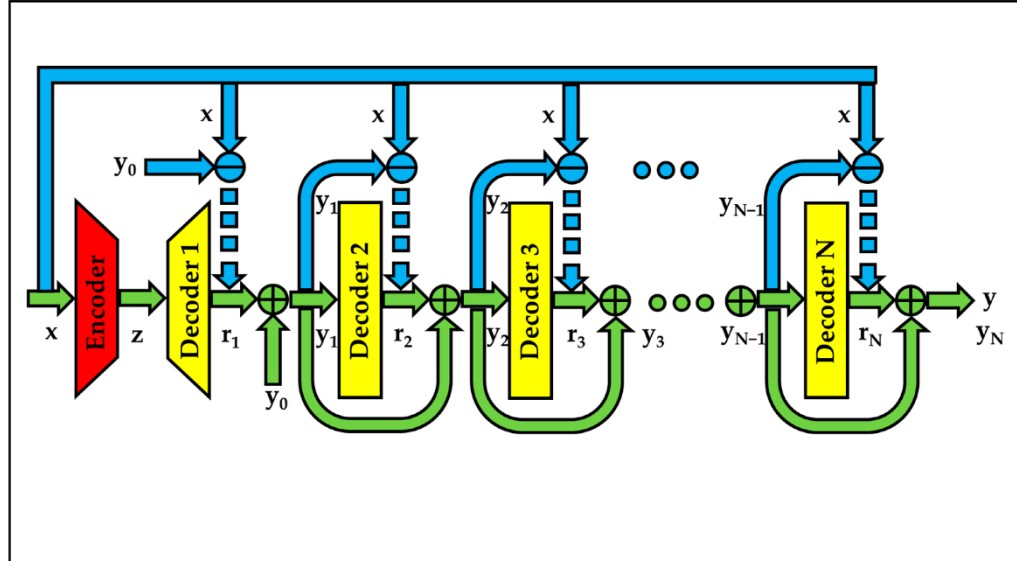

**Figure 4.** The architecture of residual cascade decoders-based auto-encoders.

For the purpose of gradually and serially training residual cascade decoders-based auto-encoders, the optimization problem in Equation (6) can be partitioned into the following suboptimization problems:

$$
\begin{aligned}
(\boldsymbol{\theta}_1; \mathbf{z}; \mathbf{r}_1; \mathbf{y}_1) &= \underset{\boldsymbol{\theta}_1; \mathbf{z}; \mathbf{r}_1; \mathbf{y}_1}{\arg\min} \|\mathbf{x} - \mathbf{y}_0 - \mathbf{r}_1\|_2^2 \\
(\boldsymbol{\theta}_2; \mathbf{r}_2; \mathbf{y}_2) &= \underset{\boldsymbol{\theta}_2; \mathbf{r}_2; \mathbf{y}_2}{\arg\min} \|\mathbf{x} - \mathbf{y}_1 - \mathbf{r}_2\|_2^2 \\
&\quad \cdots\cdots \\
(\boldsymbol{\theta}_N; \mathbf{r}_N; \mathbf{y}_N; \mathbf{y}) &= \underset{\boldsymbol{\theta}_N; \mathbf{r}_N; \mathbf{y}_N; \mathbf{y}}{\arg\min} \|\mathbf{x} - \mathbf{y}_{N-1} - \mathbf{r}_N\|_2^2 \\
\text{s.t. } &\mathbf{z} = E(\mathbf{x}); \mathbf{r}_1 = D_1(\mathbf{z}); \mathbf{r}_n = D_n(\mathbf{y}_{n-1}), n = 2, \ldots, N; \\
&C_\mathbf{z}; C_{\mathbf{y}_n}, n = 1, \ldots, N; \mathbf{y}_0 = 0; \mathbf{y}_n = \mathbf{r}_n + \mathbf{y}_{n-1}, n = 1, \ldots, N; \mathbf{y} = \mathbf{y}_N
\end{aligned}
\tag{7}
$$

The effectiveness of the residual cascade that decodes for image reconstruction can be proven as follows:

$$
\begin{aligned}
\mathbf{y}_n &= \mathbf{r}_n + \mathbf{y}_{n-1}, n = 1, \cdots, N \\
&\Rightarrow \mathbf{x} - \mathbf{y}_n = (\mathbf{x} - \mathbf{y}_{n-1}) - \mathbf{r}_n \\
\mathbf{r}_n \to (\mathbf{x} - \mathbf{y}_{n-1}) &\Rightarrow \mu = \lim_{\mathbf{r}_n \to (\mathbf{x}-\mathbf{y}_{n-1})} \frac{\mathbf{r}_n}{(\mathbf{x}-\mathbf{y}_{n-1})} \to 1 \\
\mathbf{r}_n \to (\mathbf{x} - \mathbf{y}_{n-1}) &\overset{\mu \to 1, \varepsilon \to 0}{\Rightarrow} \mathbf{r}_n = \mu(\mathbf{x} - \mathbf{y}_{n-1}) + \varepsilon, \\
\Rightarrow \mathbf{x} - \mathbf{y}_n = (\mathbf{x} - \mathbf{y}_{n-1}) &- \mu(\mathbf{x} - \mathbf{y}_{n-1}) - \varepsilon = (1 - \mu)(\mathbf{x} - \mathbf{y}_{n-1}) - \varepsilon \\
&\Rightarrow \|\mathbf{x} - \mathbf{y}_n\|_2 = \|(1 - \mu)(\mathbf{x} - \mathbf{y}_{n-1}) - \varepsilon\|_2 \\
&\Rightarrow \|\mathbf{x} - \mathbf{y}_n\|_2 < \|(1 - \mu)(\mathbf{x} - \mathbf{y}_{n-1})\|_2 + \|\varepsilon\|_2 \\
\Rightarrow \|\mathbf{x} - \mathbf{y}_n\|_2 &\overset{\varepsilon \to 0}{<} \|(1 - \mu)(\mathbf{x} - \mathbf{y}_{n-1})\|_2 = |1 - \mu| \|(\mathbf{x} - \mathbf{y}_{n-1})\|_2 \\
&\Rightarrow \|\mathbf{x} - \mathbf{y}_n\|_2 \overset{\mu \to 1}{<} 1 \cdot \|(\mathbf{x} - \mathbf{y}_{n-1})\|_2 = \|(\mathbf{x} - \mathbf{y}_{n-1})\|_2 \\
&\Rightarrow \|\mathbf{x} - \mathbf{y}_N\|_2 < \|\mathbf{x} - \mathbf{y}_{N-1}\|_2 < \cdots < \|\mathbf{x} - \mathbf{y}_2\|_2 < \|\mathbf{x} - \mathbf{y}_1\|_2
\end{aligned}
\tag{8}
$$

where $\mathbf{r}_n$ is close to $(\mathbf{x} - \mathbf{y}_{n-1})$ in the training phase in Equation (6) and Figure 4; $\mathbf{r}_n$ is the summation of a scaled $(\mathbf{x} - \mathbf{y}_{n-1})$ and a small error; u is a scale coefficient which is approximate to 1; $\varepsilon$ is an error vector which is approximate to 0; reconstruction error decreases when the total number of decoders increases.

### 3.3.2. Adversarial Cascade Decoders-Based Auto-Encoders

The architecture of adversarial cascade decoders-based auto-encoders (ACDAE) is displayed in Figure 5. The blue flow line represents the training phase, and the green flow represents both phases of training and testing. Each decoder is an adversarial module.

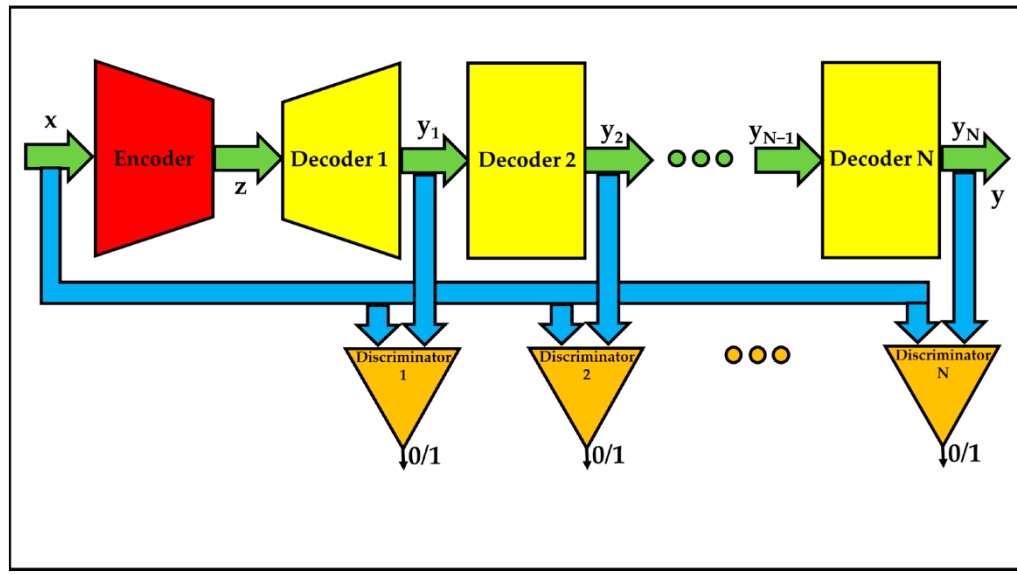

**Figure 5.** The architecture of adversarial cascade decoders-based auto-encoders.

The reconstruction data can be solved by the following optimization problem:

$$(\theta; \mathbf{z}; \mathbf{y}_1, \cdots, \mathbf{y}_N; \mathbf{y}) = \underset{E;D}{\text{argmin}} \underset{DC_1,\cdots,DC_N}{\text{max}} \sum_{n=1}^{N} (\alpha_n M(\ln(DC_n(\mathbf{x}))) + \beta_n M(\ln(1 - DC_n(\mathbf{y}_n))))$$
$$\text{s.t. } \mathbf{z} = E(\mathbf{x}); \mathbf{y}_1 = D_1(\mathbf{z}); \mathbf{y}_n = D_n(\mathbf{y}_{n-1}), n = 2, \dots, N;$$
$$C_{\mathbf{z}}; \sum_{n=1}^{N} \|\mathbf{y}_n - \mathbf{x}\|_2^2 < \varepsilon, n = 1, \dots, N;$$
$$\mathbf{y} = \mathbf{y}_N \tag{9}$$

where the variables are described as follows:

$DC_n$ is the nth discriminator;

$\alpha_n$ is a constant;

$\beta_n$ is a constant;

$\varepsilon$ is a small positive constant;

M is the mean operator.

For the sake of gradually and serially training adversarial cascade decoders-based auto-encoders, the optimization problem in Equation (9) can be divided into the following sub optimization problems:

$$(\theta_1; \mathbf{z}; \mathbf{y}_1) = \underset{E;D_1}{\text{argmin}} \underset{DC_1}{\text{max}} (\alpha_1 M(\ln(DC_1(\mathbf{x}))) + \beta_1 M(\ln(1 - DC_1(\mathbf{y}_1))))$$
$$(\theta_2; \mathbf{y}_2) = \underset{D_2}{\text{argmin}} \underset{DC_2}{\text{max}} (\alpha_2 M(\ln(DC_2(\mathbf{x}))) + \beta_2 M(\ln(1 - DC_2(\mathbf{y}_2))))$$
$$......$$
$$(\theta_N; \mathbf{y}_N; \mathbf{y}) = \underset{D_N}{\text{argmin}} \underset{DC_N}{\text{max}} (\alpha_N M(\ln(DC_N(\mathbf{x}))) + \beta_N M(\ln(1 - DC_N(\mathbf{y}_N)))) \tag{10}$$
$$\text{s.t. } \mathbf{z} = E(\mathbf{x}); \mathbf{y}_1 = D_1(\mathbf{z}); \mathbf{y}_n = D_n(\mathbf{y}_{n-1}), n = 2, \dots, N;$$
$$C_{\mathbf{z}}; \sum_{n=1}^{N} \|\mathbf{y}_n - \mathbf{x}\|_2^2 < \varepsilon, n = 1, \dots, N;$$
$$\mathbf{y} = \mathbf{y}_N$$

### 3.3.3. Residual-Adversarial Cascade Decoders-Based Auto-Encoders

The framework of residual-adversarial cascade decoders-based auto-encoders (RAC-DAE) is shown in Figure 6. The blue signal line denotes the training phase, and the green signal line denotes both phases of training and testing. Each decoder is a residual-adversarial module.

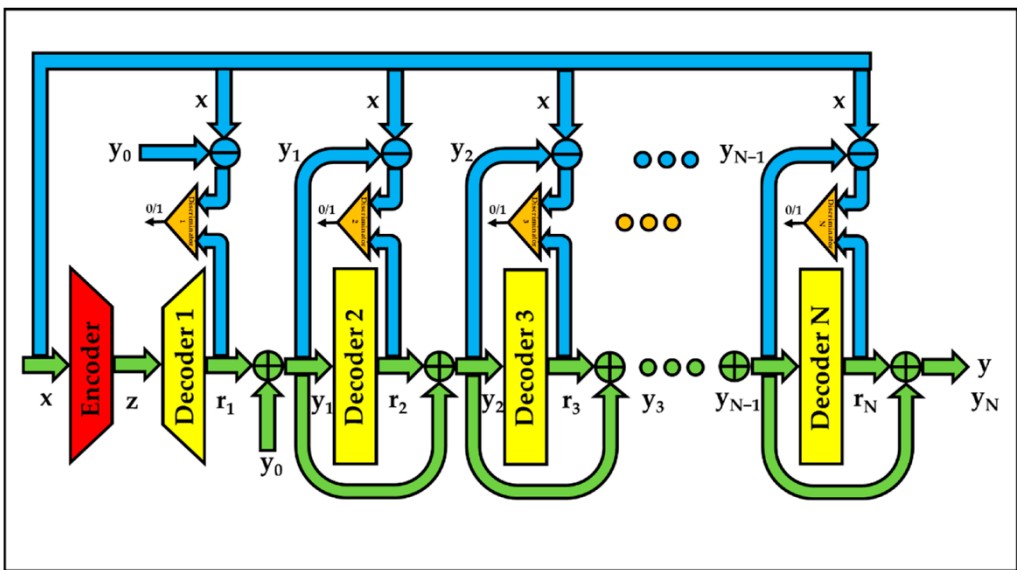

**Figure 6.** The architecture of residual-adversarial cascade decoders-based auto-encoders.

The recovery data can be resolved by the following minimization-maximization problem:

$$
\begin{aligned}
(\theta; \mathbf{z}; \mathbf{r}_1, \cdots, \mathbf{r}_N; \mathbf{y}_1, \cdots, \mathbf{y}_N; \mathbf{y}) = \underset{E;D}{\arg\min} \underset{DC_1, \cdots, DC_N}{\max} \sum_{n=1}^{N} \left( \alpha_n M \left( \ln \left( DC_n \left( \mathbf{x} - \mathbf{y}_{n-1} \right) \right) \right) + \beta_n M \left( \ln \left( 1 - DC_n(\mathbf{r}_n) \right) \right) \right) \\
\text{s.t. } \mathbf{z} = E(\mathbf{x}); \mathbf{r}_1 = D_1(\mathbf{z}); \mathbf{r}_n = D_n(\mathbf{y}_{n-1}), n = 2, \ldots, N; \\
C_{\mathbf{z}}; \sum_{n=1}^{N} \|\mathbf{x} - \mathbf{y}_{n-1} - \mathbf{r}_n\|_2^2 < \varepsilon, n = 1, \ldots, N; \\
\mathbf{y}_0 = 0; \mathbf{y}_n = \mathbf{y}_{n-1} + \mathbf{r}_n, n = 1, \ldots, N; \mathbf{y} = \mathbf{y}_N
\end{aligned}
\tag{11}
$$

For the purpose of gradually and serially training residual adversarial cascade decoders-based auto-encoders, the optimization problem in Equation (11) can be divided into the following suboptimization problems:

$$
\begin{aligned}
(\theta_1; \mathbf{z}; \mathbf{r}_1; \mathbf{y}_1) = \underset{E;D_1}{\arg\min}\underset{DC_1}{\max}(\alpha_1 M(\ln(DC_1(\mathbf{x} - \mathbf{y}_0))) + \beta_1 M(\ln(1 - DC_1(\mathbf{r}_1)))) \\
(\theta_2; \mathbf{r}_2; \mathbf{y}_2) = \underset{D_2}{\arg\min}\underset{DC_2}{\max}(\alpha_2 M(\ln(DC_2(\mathbf{x} - \mathbf{y}_1))) + \beta_2 M(\ln(1 - DC_2(\mathbf{r}_2)))) \\
\cdots\cdots \\
(\theta_N; \mathbf{r}_N; \mathbf{y}_N; \mathbf{y}) = \underset{D_N}{\arg\min}\underset{DC_N}{\max}(\alpha_N M(\ln(DC_N(\mathbf{x} - \mathbf{y}_{N-1}))) + \beta_N M(\ln(1 - DC_N(\mathbf{r}_N)))) \\
\text{s.t. } \mathbf{z} = E(\mathbf{x}); \mathbf{r}_1 = D_1(\mathbf{z}); \mathbf{r}_n = D_n(\mathbf{y}_{n-1}), n = 2, \ldots, N; \\
C_{\mathbf{z}}; \sum_{n=1}^{N} \|\mathbf{x} - \mathbf{y}_{n-1} - \mathbf{r}_n\|_2^2 < \varepsilon, n = 1, \ldots, N; \\
\mathbf{y}_0 = 0; \mathbf{y}_n = \mathbf{y}_{n-1} + \mathbf{r}_n, n = 1, \ldots, N; \mathbf{y} = \mathbf{y}_N
\end{aligned}
\tag{12}
$$

### 3.4. Adversarial Auto-Encoders

3.4.1. Reminiscence of Classical Adversarial Auto-Encoders

The infrastructure of the classical adversarial auto-encoders is exhibited in Figure 7. The blue signal flow is for the training phase, and the green signal flow is for both phases of training and testing. AAE are the combination of auto-encoders and adversarial learning. Alireza Makhzani et al. proposed the AAE, utilized the encoder unit of auto-encoders

as generator and added an independent discriminator, employed adversarial learning in latent space and let the hidden variable satisfy a given distribution, and finally achieved better performance in data reconstruction [7]. Compared with the classical auto-encoders, AAE infrastructure holds an extra discriminator, which makes the output of the encoder maximally approach a given distribution. The infrastructure can be expressed by the following equations:

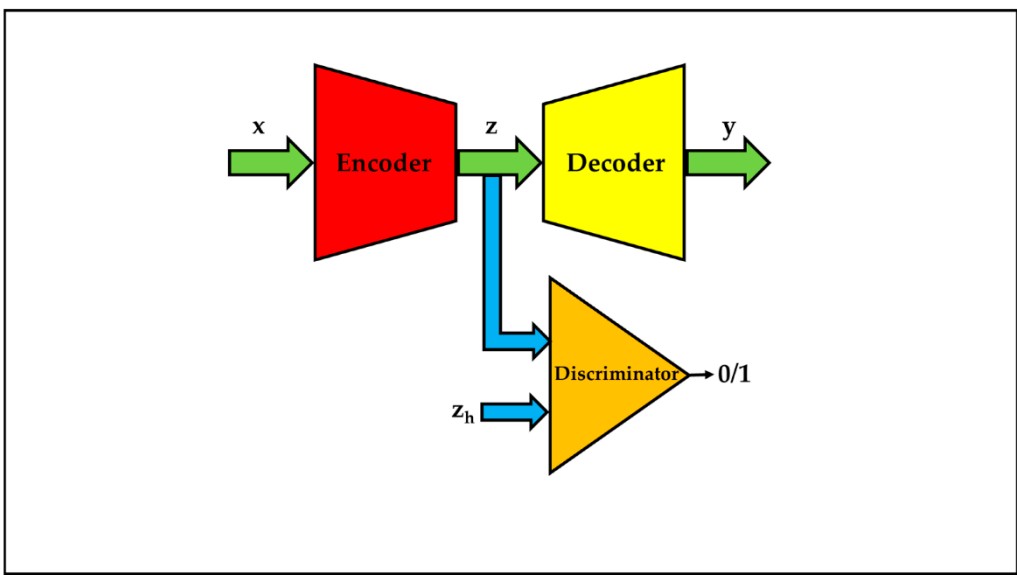

**Figure 7.** The architecture of classical adversarial auto-encoders.

$$
\begin{aligned}
\mathbf{z} &= E(\mathbf{x}) \\
\mathbf{y} &= D(\mathbf{z}) \\
DC(\mathbf{z}_h) &= 1, DC(\mathbf{z}) = 0
\end{aligned}
\tag{13}
$$

where the terms are defined as follows:

$\mathbf{z}_h$ is the variable related to $\mathbf{z}$ which satisfies a given distribution;
DC is the discriminator.

The reestablishment data can be resolved by the following minimization-maximization problem:

$$
(\theta, \mathbf{z}, \mathbf{y}) = \underset{E,D}{\arg\min}\underset{DC}{\max}(\alpha M(\ln(DC(\mathbf{z}_h))) + \beta M(\ln(1 - DC(\mathbf{z}))))
$$
$$
\text{s.t. } \mathbf{z} = E(\mathbf{x}), \mathbf{y} = D(\mathbf{z}), \|\mathbf{y} - \mathbf{x}\|_2^2 < \varepsilon, C_y
\tag{14}
$$

### 3.4.2. Proposed Cascade Decoders-Based Adversarial Auto-Encoders

The architecture of the proposed cascade decoders-based adversarial auto-encoders (CDAAE) is illustrated in Figure 8. The blue flow line represents the training phase, and the green flow line represents both phases of training and testing. Compared with the cascade decoders-based auto-encoders, the proposed architecture has an extra discriminator, which makes the output of the encoder maximally approximate to a known distribution. The architecture can be described by the following formulas:

$$
\begin{aligned}
\mathbf{z} &= E(\mathbf{x}) \\
\mathbf{y}_n &= \begin{cases} D_1(\mathbf{z}), n = 1 \\ D_n(\mathbf{y}_{n-1}), n = 2, \ldots, N \end{cases} \\
DC(\mathbf{z}_h) &= 1, DC(\mathbf{z}) = 0
\end{aligned}
\tag{15}
$$

The restoration data can be resolved by the following optimization problem:

$$(\boldsymbol{\theta}; \mathbf{z}; \mathbf{y}_1, \cdots, \mathbf{y}_N) = \arg\min_{E;D_1,\cdots,D_N}\max_{DC}(\alpha M(\ln(DC(\mathbf{z}_h))) + \beta M(\ln(1 - DC(\mathbf{z}))))$$
$$\text{s.t. } \mathbf{z} = E(\mathbf{x}); \mathbf{y}_1 = D_1(\mathbf{z}); \mathbf{y}_n = D_n(\mathbf{y}_{n-1}), n = 2, \ldots, N;$$
$$\sum_{n=1}^{N}\|\mathbf{y}_n - \mathbf{x}\|_2^2 < \varepsilon; C_{y_n}, n = 1, \ldots, N \tag{16}$$

For the purpose of gradually and serially training cascade decoders-based adversarial auto-encoders, the optimization problem in Equation (16) can be partitioned into the following suboptimization problems:

$$(\boldsymbol{\theta}_1; \mathbf{z}; \mathbf{y}_1) = \underset{E;D_1}{\arg\min}\max_{DC}(\alpha M(\ln(DC(\mathbf{z}_h))) + \beta M\ln(1 - DC(\mathbf{z})))$$
$$(\boldsymbol{\theta}_2; \mathbf{y}_2) = \underset{D_2}{\arg\min}\|\mathbf{y}_2 - \mathbf{x}\|_2^2$$
$$\cdots\cdots$$
$$(\boldsymbol{\theta}_N; \mathbf{y}_N) = \underset{D_N}{\arg\min}\|\mathbf{y}_N - \mathbf{x}\|_2^2$$
$$\text{s.t. } \mathbf{z} = E(\mathbf{x}); \mathbf{y}_1 = D_1(\mathbf{z}); \mathbf{y}_n = D_n(\mathbf{y}_{n-1}), n = 2, \ldots, N;$$
$$\|\mathbf{y}_1 - \mathbf{x}\|_2^2 < \varepsilon; C_{y_n}, n = 1, \ldots, N \tag{17}$$

The architecture in Figure 8 represents general cascade decoders-based adversarial auto-encoders (GCDAAE), and it can be easily be expanded to residual cascade decoders-based adversarial auto-encoders (RCDAAE), adversarial cascade decoders-based adversarial auto-encoders (ACDAAE), and residual-adversarial cascade decoders-based adversarial auto-encoders (RACDAAE).

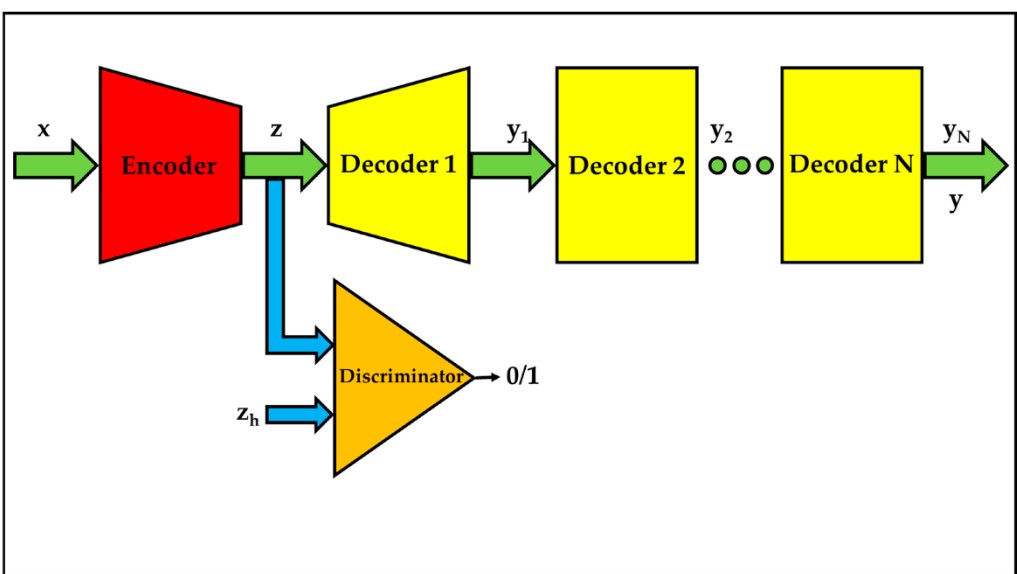

**Figure 8.** The architecture of cascade decoders-based adversarial auto-encoders.

*3.5. Variational Auto-Encoders*

3.5.1. Remembrance of Classical Variational Auto-Encoders

The framework of classical variational auto-encoders is shown in Figure 9 [4]. The blue signal line denotes the training phase, and the green signal line denotes both phases of training and testing. It can be resolved by the following optimization problem:

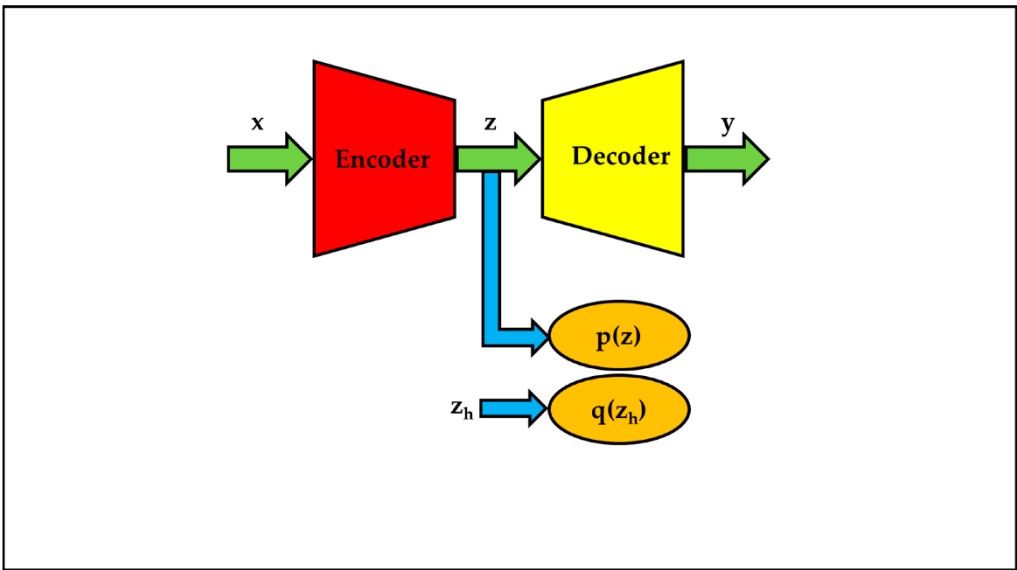

**Figure 9.** The architecture of classical variational auto-encoders.

$$(\theta, \mathbf{z}, \mathbf{y}) = \underset{\theta, \mathbf{z}, \mathbf{y}}{\mathrm{argmin}}\left(\alpha KL(q(\mathbf{z}_h)||p(\mathbf{z})) + \beta\|\mathbf{y} - \mathbf{x}\|_2^2\right)$$
$$= \underset{\theta, \mathbf{z}, \mathbf{y}}{\mathrm{argmin}}\left(\alpha\sum_{\mathbf{z}_h} q(\mathbf{z}_h)\log\frac{q(\mathbf{z}_h)}{p(\mathbf{z})} + \beta\|\mathbf{y} - \mathbf{x}\|_2^2\right) \qquad (18)$$
$$\text{s.t. } \mathbf{z} = E(\mathbf{x}), \mathbf{y} = D(\mathbf{z}), C_y$$

where the terms are defined as follows:

$KL(\cdot)$ is the Kullback–Leibler divergence;
$q(\mathbf{z}_h)$ is the given distribution of $\mathbf{z}_h$;
$p(\mathbf{z})$ is the distribution of $\mathbf{z}$.

### 3.5.2. Proposed Cascade Decoders-Based Variational Auto-Encoders

The proposed infrastructure of cascade decoders-based variational auto-encoders is shown in Figure 10. The blue signal flow is for the training phase, and the green signal flow is for both phases of training and testing. It can be resolved by the following optimization problem:

$$(\theta; \mathbf{z}; \mathbf{y}_1, \cdots, \mathbf{y}_N) = \underset{\theta; \mathbf{z}; \mathbf{y}_1, \cdots, \mathbf{y}_N}{\mathrm{argmin}}\left(\alpha KL(q(\mathbf{z}_h)||p(\mathbf{z})) + \beta\sum_{n=1}^{N}\|\mathbf{y}_n - \mathbf{x}\|_2^2\right) \qquad (19)$$
$$\text{s.t. } \mathbf{z} = E(\mathbf{x}); \mathbf{y}_1 = D_1(\mathbf{z}); \mathbf{y}_n = D_n(\mathbf{y}_{n-1}), n = 2, \ldots, N; C_{y_n}, n = 1, \ldots, N$$

For the sake of gradually and serially training cascade decoders-based auto-encoders, the optimization problem in Equation (19) can be divided into the following suboptimization problems:

$$(\theta_1; \mathbf{z}; \mathbf{y}_1) = \underset{\theta_1; \mathbf{z}; \mathbf{y}_1}{\mathrm{argmin}}\left(\alpha KL(q(\mathbf{z}_h)||p(\mathbf{z})) + \beta\|\mathbf{y}_1 - \mathbf{x}\|_2^2\right)$$
$$(\theta_2; \mathbf{y}_2) = \underset{\theta_2; \mathbf{y}_2}{\mathrm{argmin}}\|\mathbf{y}_2 - \mathbf{x}\|_2^2$$
$$\cdots\cdots \qquad (20)$$
$$(\theta_N; \mathbf{y}_N) = \underset{\theta_N; \mathbf{y}_N}{\mathrm{argmin}}\|\mathbf{y}_N - \mathbf{x}\|_2^2$$
$$\text{s.t. } \mathbf{z} = E(\mathbf{x}); \mathbf{y}_1 = D_1(\mathbf{z}); \mathbf{y}_n = D_n(\mathbf{y}_{n-1}), n = 2, \ldots, N; C_{y_n}, n = 1, \ldots, N$$

The infrastructure in Figure 10 represents general cascade decoders-based variational auto-encoders (GCDVAE), and it can be easily be extended to residual cascade decoders-

based variational auto-encoders (RCDVAE), adversarial cascade decoders-based variational auto-encoders (ACDVAE), and residual-adversarial cascade decoders-based variational auto-encoders (RACDVAE).

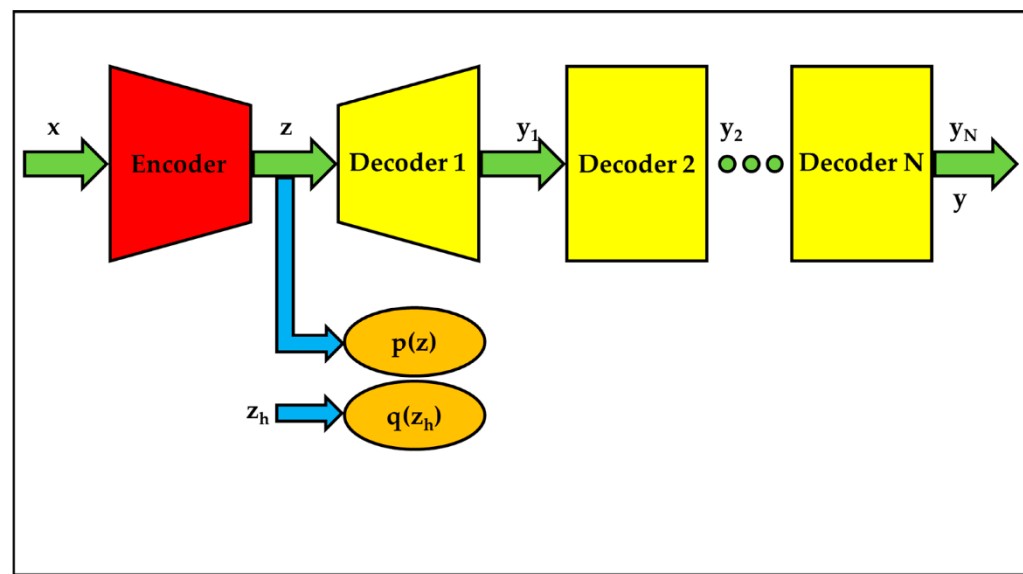

**Figure 10.** The architecture of cascade decoders-based variational auto-encoders.

*3.6. Wasserstein Auto-Encoders*

3.6.1. Recollection of Classical Wasserstein Auto-Encoders

The classical Wasserstein auto-encoders can be resolved by the following optimization problem [20]:

$$(\theta, \mathbf{z}, \mathbf{y}) = \underset{\theta, \mathbf{z}, \mathbf{y}}{\arg\min} \big( \alpha W_z(p(\mathbf{z}), q(\mathbf{z}_h)) + \beta W_y(\mathbf{y}, \mathbf{x}) \big) \tag{21}$$
$$\text{s.t. } \mathbf{z} = E(\mathbf{x}), \mathbf{y} = D(\mathbf{z}), C_y$$

where the variables are defined as follows:

$W_z$ is the regularizer between distribution $p(\mathbf{z})$ and $q(\mathbf{z}_h)$;
$W_y$ is the reconstruction cost.

3.6.2. Proposed Cascade Decoders-Based Wasserstein Auto-Encoders

The proposed cascade decoders-based Wasserstein auto-encoders can be resolved by the following optimization problem:

$$(\theta; \mathbf{z}; \mathbf{y}_1, \cdots, \mathbf{y}_N) = \underset{\theta; \mathbf{z}; \mathbf{y}_1, \cdots, \mathbf{y}_N}{\arg\min} \left( \alpha W_z(p(\mathbf{z}), q(\mathbf{z}_h)) + \beta \sum_{n=1}^{N} W_y(\mathbf{y}_n, \mathbf{x}) \right) \tag{22}$$
$$\text{s.t. } \mathbf{z} = E(\mathbf{x}); \mathbf{y}_1 = D_1(\mathbf{z}); \mathbf{y}_n = D_n(\mathbf{y}_{n-1}), n = 2, \ldots, N; C_{y_n}, n = 1, \ldots, N$$

For the purpose of gradually and serially training cascade decoders-based auto-encoders, the optimization problem in Equation (22) can be divided into the following suboptimization problems:

$$(\theta_1; \mathbf{z}; \mathbf{y}_1) = \underset{\theta_1; \mathbf{z}; \mathbf{y}_1}{\arg\min} \big( \alpha W_z(p(\mathbf{z}), q(\mathbf{z}_h)) + \beta W_y(\mathbf{y}_1, \mathbf{x}) \big)$$
$$(\theta_2; \mathbf{y}_2) = \underset{\theta_2; \mathbf{y}_2}{\arg\min} W_y(\mathbf{y}_2, \mathbf{x})$$
$$\cdots \cdots \tag{23}$$
$$(\theta_N; \mathbf{y}_N) = \underset{\theta_N; \mathbf{y}_N}{\arg\min} W_y(\mathbf{y}_N, \mathbf{x})$$
$$\text{s.t. } \mathbf{z} = E(\mathbf{x}); \mathbf{y}_1 = D_1(\mathbf{z}); \mathbf{y}_n = D_n(\mathbf{y}_{n-1}), n = 2, \ldots, N; C_{y_n}, n = 1, \ldots, N$$

The aforementioned architecture represents general cascade decoders-based Wasserstein auto-encoders (GCDWAE), and it can be easily be expanded to residual cascade decoders-based Wasserstein auto-encoders (RCDWAE), adversarial cascade decoders-based Wasserstein auto-encoders (ACDWAE), and residual-adversarial cascade decoders-based Wasserstein auto-encoders (RACDWAE).

### 3.7. Pseudocodes of Cascade Decoders-Based Auto-Encoders

The pseudo codes of the proposed cascade decoders-based auto-encoders are shown in Algorithm 1.

---

**Algorithm 1:** The pseudo codes of cascade decoders-based auto-encoders.

---

**Input: x**: the training data
    I: the total number of iteration
    N: the total number of sub minimization problem
**Initialization:** i =1
**Training:**
    While i <= I
        i++
        n = 1
        While n <= N
            n++
            resolve the nth sub problem in Equations (5), (7), (10), (12), (17), (20) or (23)
**Output:**
    θ: the parameters of deep neural networks
    **z:** the representations of hidden space
    **$y_1, \ldots, y_N$:** the output of cascade decoders
    **y:** the output of the last decoder

---

## 4. Experiments

### 4.1. Experimental Data Sets

The purpose of the simulation experiments is to compare the data reconstruction performance of the proposed cascade decoders-based auto-encoders and the classical auto-encoders.

Four data sets are utilized to evaluate algorithm performance [29–32]. The mixed national institute of standards and technology (MNIST) data set has 10 classes of handwritten digit images [29]; the extending MNIST (EMNIST) data set holds 6 subcategories of handwritten digit and letter images [30]; the fashion-MNIST (FMNIST) data set possesses 10 classes of fashion product images [31]; and the medical MNIST (MMNIST) data set owns 10 subcategories of medical images [32]. The image size is $28 \times 28$. All color images are converted into gray images. In order to reduce the computational load, small resolution images and gray images are chosen. Certainly, if the computational capability is ensured, the proposed methods can be easily and directly utilized on large resolution images, the components of color images or their sub-patches. A large image can be divided into small patches. In traditional image compression methods, the size of image patch for compression is $8 \times 8$. Therefore, the proposed methods can be used for each image block. In brief, large image size will not degrade the performance of the proposed methods from the viewpoint of small image patches. For the convenience of training and testing deep neural networks, each pixel value is normalized from range [0, 255] to range [−1, +1] in the phase of pre-processing, and is re-scaled back to range [0, 255] in the phase of post-processing. The numbers of classes and samples in the four data sets are enumerated in Table 3. The sample images of the four data sets are illustrated in Figure 11. From top to bottom, there are images of MNIST digits, EMINST digits, EMNIST letters, FMNIST goods, MMNIST breast, chest, derma, optical coherence tomography (OCT), axial organ, coronal organ, sagittal organ, pathology, pneumonia, and retina.

**Table 3.** Class and sample numbers of experimental data sets.

| Data Set | | Class Number | Sample Number | |
|---|---|---|---|---|
| | | | Training | Testing |
| MNIST | | 10 | 60,000 | 10,000 |
| EMINST | digits | 10 | 240,000 | 40,000 |
| | letters | 26 | 124,800 | 20,800 |
| | balanced | 47 | 112,800 | 18,800 |
| | bymerge | 47 | 697,932 | 116,323 |
| | byclass | 62 | 697,932 | 116,323 |
| FMNIST | | 10 | 60,000 | 10,000 |
| MMNIST | breast | 2 | 546 | 156 |
| | chest | 2 | 78,468 | 22,433 |
| | derma | 7 | 7007 | 2005 |
| | OCT | 4 | 97,477 | 1000 |
| | axial organ | 11 | 34,581 | 17,778 |
| | coronal organ | 11 | 13,000 | 8268 |
| | sagittal organ | 11 | 13,940 | 8829 |
| | pathology | 9 | 89,996 | 7180 |
| | pneumonia | 2 | 4708 | 624 |
| | retina | 5 | 1080 | 400 |

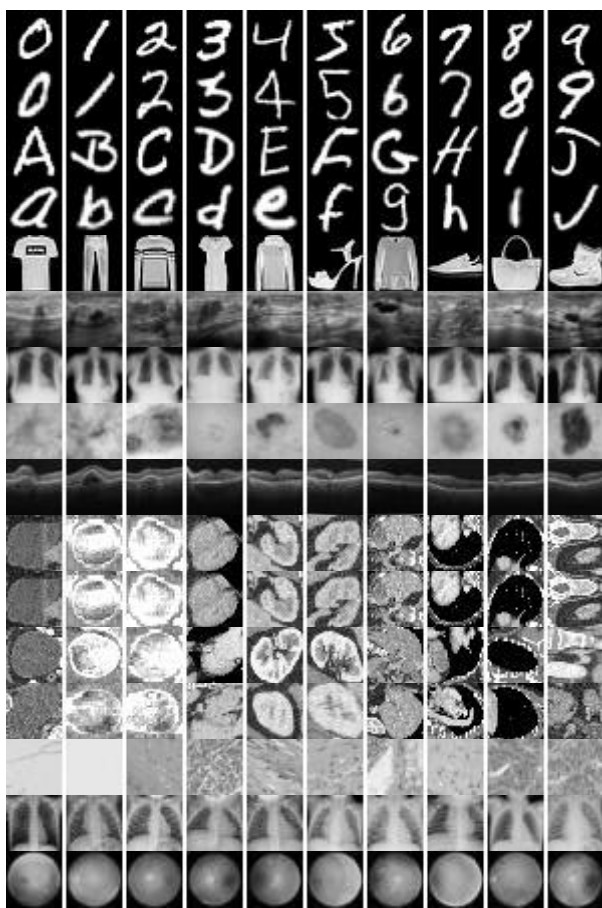

**Figure 11.** Sample images of experimental data sets.

*4.2. Experimental Conditions*

The experimental software platform is MATLAB 2020b on Windows 10 or Linux. For the small data sets, MNIST, FMNIST, and MMNIST, the experimental hardware platform is a laptop with a 2.6 GHz dual-core processor and 8 GB memory; For the large data set,

EMNIST, the experimental hardware platform is a super computer with high-speed GPUs and substantial memory.

The components of auto-encoders are made up of fully-connected (FC) layers, leaky rectified linear unit (LRELU) layers, hyperbolic tangent (Tanh) layers, Sigmoid layers, etc. In order to reduce the calculation complexity, the convolutional (CONV) layer is not utilized.

The composition of the encoder is shown in Figure 12, which consists of input, FC, LRELU, and hidden layers.

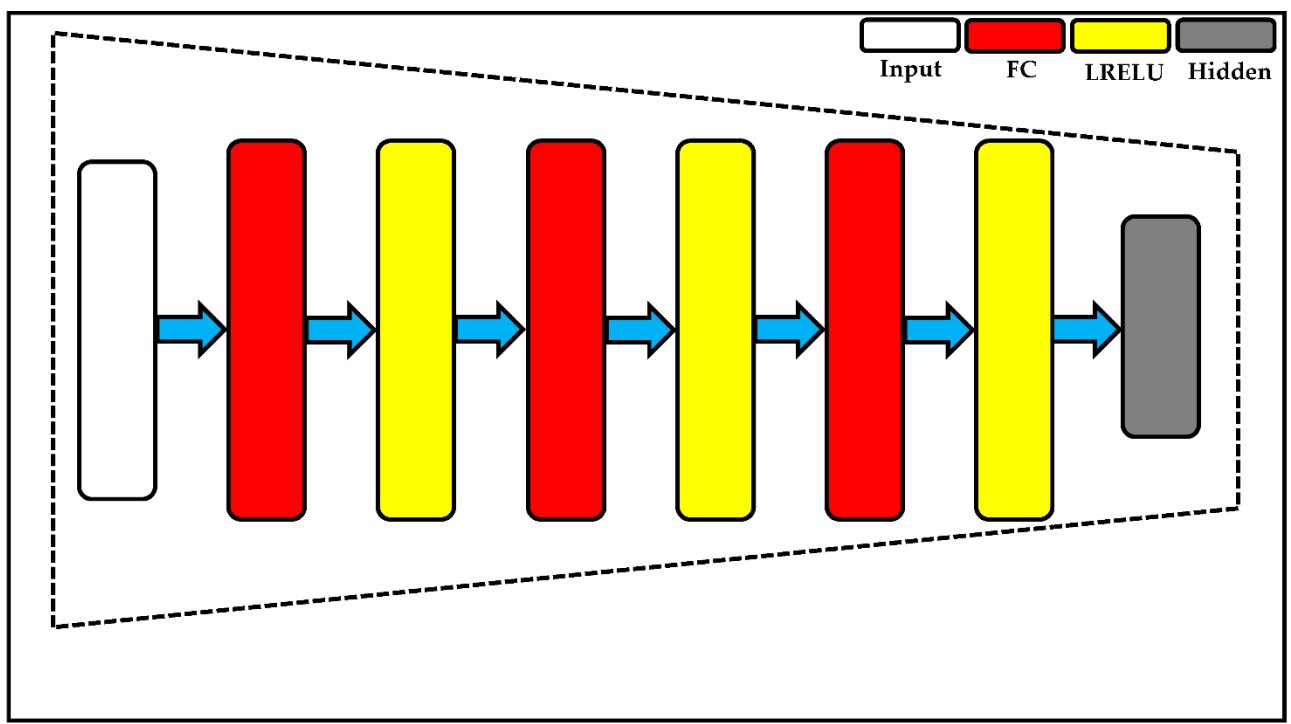

**Figure 12.** Composition of the encoder.

The constitution of the decoder is illustrated in Figure 13, which consists of input, FC, LRELU, Tanh and output layers. The input layer can be the hidden layer for the first decoder, and can be the output layer of the preceding decoder for the latter decoders. The dashed line shows the two situations.

The organization of the discriminator is demonstrated in Figure 14, which comprises input, FC, LRELU, Sigmoid, and output layers. The input layer can be the hidden layer and can be the output of each decoder. The dashed line indicates the two cases.

The deep learning parameters, such as image size, latent dimension, decoder number, batch size, learning rate, and iteration epoch, are summarized in Table 4.

**Table 4.** The deep learning parameters.

| Parameter Name | Parameter Value |
| --- | --- |
| Image size | $28 \times 28 \times 1$ |
| Latent Dimension | 30 |
| Decoder Number | 3 |
| Batch size | 100 |
| Learning Rate | 0.0002 |
| Iteration Epoch | 100 |

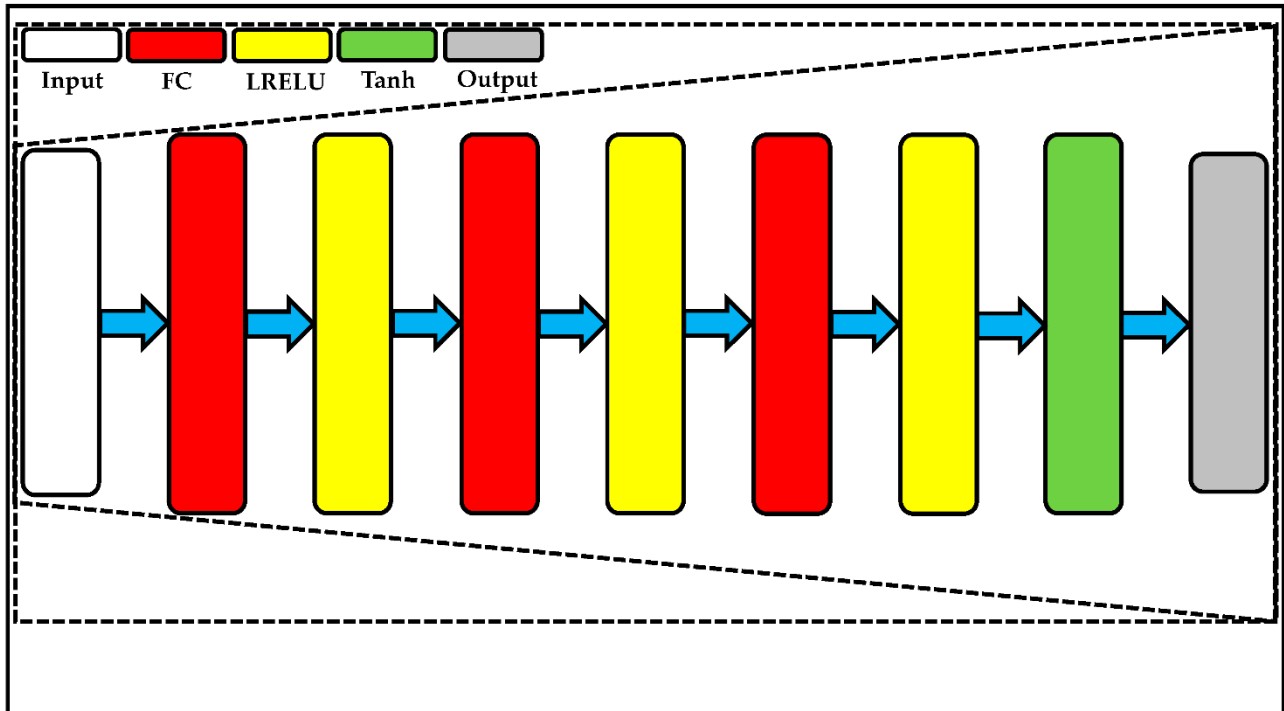

**Figure 13.** Composition of the decoder.

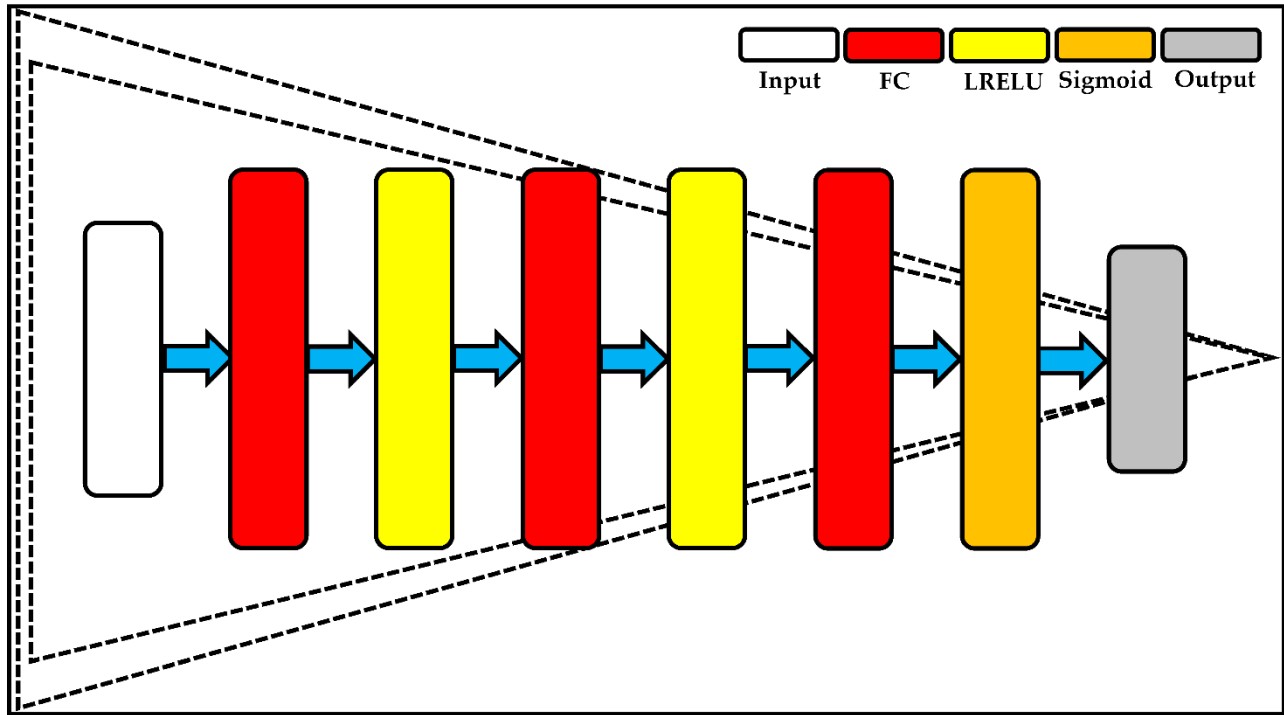

**Figure 14.** Composition of the discriminator.

*4.3. Experimental Results*

The experimental results of the proposed and classical algorithms on the MNIST data set, EMNIST data set, FMNIST data set, and MMNIST data set are respectively shown in Tables 5–8. SSIM is the average structure similarity between reconstruction images and original images. ΔSIMM is the average SSIM difference between the proposed approaches

and the conventional AE approach. The experimental results are also displayed in Figure 15, where the horizontal coordinate is data sets and the vertical coordinate is ΔSIMM.

It can be found in Tables 5–8 and Figure 15 that the proposed methods, except for ACDAE and ACDAAE, are superior to the classical AE and AAE methods in the performance of image reconstruction. Therefore, it proves the correctness and effectiveness of the proposed cascade decoders-based auto-encoders for image reconstruction.

It can also be discovered in Tables 5–8 and Figure 15 that the proposed RCDAE and RACDAAE algorithms post the best recovery performance across nearly all four data sets. Hence, residual learning is very suitable for image recovery. This is owing to the fact that the residual has a smaller average and variance than the original image, which is beneficial for the deep neural network to learn relationships between the input and output.

**Table 5.** Experimental results on the MNIST data set.

| Algorithms | SSIM | ΔSSIM |
|:---:|:---:|:---:|
| AE | 0.97387 | 0.00000 |
| GCDAE | 0.97415 | 0.00028 |
| RCDAE | **0.97592** | **0.00205** |
| ACDAE | 0.97354 | −0.00033 |
| RACDAE | 0.97574 | 0.00187 |
| AAE | 0.97304 | −0.00083 |
| GCDAAE | 0.97397 | 0.00010 |
| RCDAAE | 0.97576 | 0.00189 |
| ACDAAE | 0.97381 | −0.00006 |
| RACDAAE | 0.97589 | 0.00202 |

**Table 6.** Experimental results on the EMNIST data set.

| Algorithms | SSIM ΔSSIM | | | | |
|:---:|:---:|:---:|:---:|:---:|:---:|
| | EMNIST | | | | |
| | **Digits** | **Letters** | **Balanced** | **Bymerge** | **Byclass** |
| AE | 0.98322 0.00000 | 0.97466 0.00000 | 0.97277 0.00000 | 0.98288 0.00000 | 0.98309 0.00000 |
| GCDAE | 0.98380 0.00058 | 0.97466 0.00000 | 0.97315 0.00038 | 0.98423 0.00135 | 0.98432 0.00123 |
| RCDAE | 0.98528 0.00206 | 0.97672 0.00206 | 0.97528 0.00251 | **0.98589 0.00301** | **0.98597 0.00288** |
| ACDAE | 0.98285 −0.00037 | 0.97391 −0.00075 | 0.97223 −0.00054 | 0.98300 0.00012 | 0.98314 0.00005 |
| RACDAE | 0.98517 0.00195 | 0.97654 0.00188 | 0.97570 0.00293 | 0.98433 0.00145 | 0.98517 0.00208 |
| AAE | 0.98304 −0.00018 | 0.97434 −0.00032 | 0.97291 0.00014 | 0.98291 0.00003 | 0.98284 −0.00025 |
| GCDAAE | 0.98375 0.00053 | 0.97451 −0.00015 | 0.97305 0.00028 | 0.98413 0.00125 | 0.98476 0.00167 |
| RCDAAE | 0.98534 0.00212 | 0.97659 0.00193 | 0.97546 0.00269 | 0.98586 0.00298 | 0.98592 0.00283 |
| ACDAAE | 0.98305 −0.00017 | 0.97410 −0.00056 | 0.97275 −0.00020 | 0.98329 0.00041 | 0.98340 0.00031 |
| RACDAAE | **0.98543 0.00221** | **0.97708 0.00242** | **0.97593 0.00316** | 0.98529 0.00241 | 0.98531 0.00222 |

**Table 7.** Experimental results on the FMNIST data set.

| Algorithms | SSIM | ΔSSIM |
|---|---|---|
| AE | 0.96335 | 0.00000 |
| GCDAE | 0.96463 | 0.00128 |
| RCDAE | 0.96620 | 0.00285 |
| ACDAE | 0.96329 | −0.00006 |
| RACDAE | 0.96625 | 0.00290 |
| AAE | 0.96366 | 0.00031 |
| GCDAAE | 0.96458 | 0.00123 |
| RCDAAE | 0.96630 | 0.00295 |
| ACDAAE | 0.96353 | 0.00018 |
| RACDAAE | **0.96637** | **0.00302** |

**Table 8.** Experimental results on the MMNIST data set.

| Algorithms | SSIM ΔSSIM | | | | | | | | | |
|---|---|---|---|---|---|---|---|---|---|---|
| | MedMNIST | | | | | | | | | |
| | Breast | Chest | Derma | Oct | Axial | Coronal | Sagittal | Pathology | Pneumonia | Retina |
| AE | 0.88868 0.00000 | 0.98363 0.00000 | 0.97103 0.00000 | 0.98851 0.00000 | 0.81806 0.00000 | 0.82470 0.00000 | 0.79883 0.00000 | 0.89644 0.00000 | 0.94760 0.00000 | 0.97366 0.00000 |
| GCDAE | 0.89228 0.00360 | 0.98836 0.00473 | 0.97103 0.00000 | 0.98856 0.00005 | 0.92817 0.11011 | 0.83320 0.00850 | 0.84211 0.04328 | 0.89720 0.00076 | 0.95782 0.01022 | 0.97751 0.00385 |
| RCDAE | **0.90574** **0.01706** | **0.98857** **0.00494** | **0.97396** **0.00295** | **0.98927** **0.00076** | 0.90092 0.08286 | 0.83467 0.00997 | 0.84180 0.04297 | 0.89724 0.00080 | **0.96115** **0.01355** | 0.98233 0.00867 |
| ACDAE | 0.87883 −0.00985 | 0.98682 0.00319 | 0.96834 −0.00269 | 0.98780 −0.00071 | 0.93012 0.11206 | 0.83450 0.00980 | **0.84302** **0.04419** | 0.89548 −0.00096 | 0.95469 0.00709 | 0.97892 0.00526 |
| RACDAE | 0.89908 0.01040 | 0.98543 0.00180 | 0.97028 −0.00075 | 0.98832 −0.00019 | 0.89931 0.08125 | **0.83494** **0.01024** | 0.84155 0.04272 | 0.89594 −0.00050 | 0.95784 0.01008 | 0.98067 0.00701 |
| AAE | 0.88527 −0.00341 | 0.97240 −0.01230 | 0.97052 −0.00051 | 0.98858 0.00007 | 0.81498 −0.00308 | 0.83347 0.00877 | 0.84202 0.04319 | 0.89603 −0.00041 | 0.95356 0.00596 | 0.97646 0.00280 |
| GCDAAE | 0.89139 0.00271 | 0.98847 0.00484 | 0.97054 −0.00053 | 0.98867 0.00016 | **0.93736** **0.11930** | 0.83347 0.01000 | 0.84261 0.04378 | 0.89714 0.00070 | 0.95811 0.01051 | 0.97839 0.00473 |
| RCDAAE | 0.89772 0.00094 | 0.98852 0.00489 | 0.97262 0.00159 | 0.98920 0.00069 | 0.89873 0.08607 | 0.83321 0.00851 | 0.84179 0.04296 | **0.89747** **0.00103** | 0.96076 0.01316 | **0.98261** **0.00895** |
| ACDAAE | 0.88895 0.00027 | 0.98673 0.00310 | 0.96987 −0.00116 | 0.98769 −0.00082 | 0.93286 0.11480 | 0.83421 0.00951 | 0.84129 0.04246 | 0.89695 0.00051 | 0.95389 0.00629 | 0.97942 0.00576 |
| RACDAAE | 0.90113 0.01245 | 0.98831 0.00468 | 0.97062 −0.00041 | 0.98865 0.00014 | 0.90580 0.08702 | 0.83380 −0.00090 | 0.83975 0.04092 | 0.89728 0.00084 | 0.95720 0.00510 | 0.98117 0.00751 |

It can further be observed in Tables 5–8 and Figure 15 that the proposed ACDAE and ACDAAE algorithms yield some minus ΔSIMM across the four data sets. Thus, in line with the predictions of this paper, pure adversarial learning is unsuitable for image re-establishment. However, a combination of residual learning and adversarial learning, such as the aforementioned RACDAAE, can obtain high re-establishment performance.

It can additionally be found in Tables 5–8 and Figure 15 that the AAE algorithm possesses some minus ΔSIMM across the four data sets. Therefore, consistent with the circumstances of this article, pure AAE cannot outperform AE in image reconstitution. Nevertheless, a combination of residual learning and adversarial learning, such as aforementioned RACDAAE, can produce high reconstitution performance.

Finally, it can be found in Tables 5–8 and Figure 15 that SSIM differences for MMNIST-axial and MMNIST-sagittal are substantially higher than for other data sets. The reason for this may be that these training and testing samples are more similar than other data sets.

In order to clearly compare the reconstruction performance between the proposed algorithms and the classical algorithms, the reconstruction images are illustrated in Figures 16–25.

Since the proposed RCDAE algorithm owns the best performance, it is taken as an example.

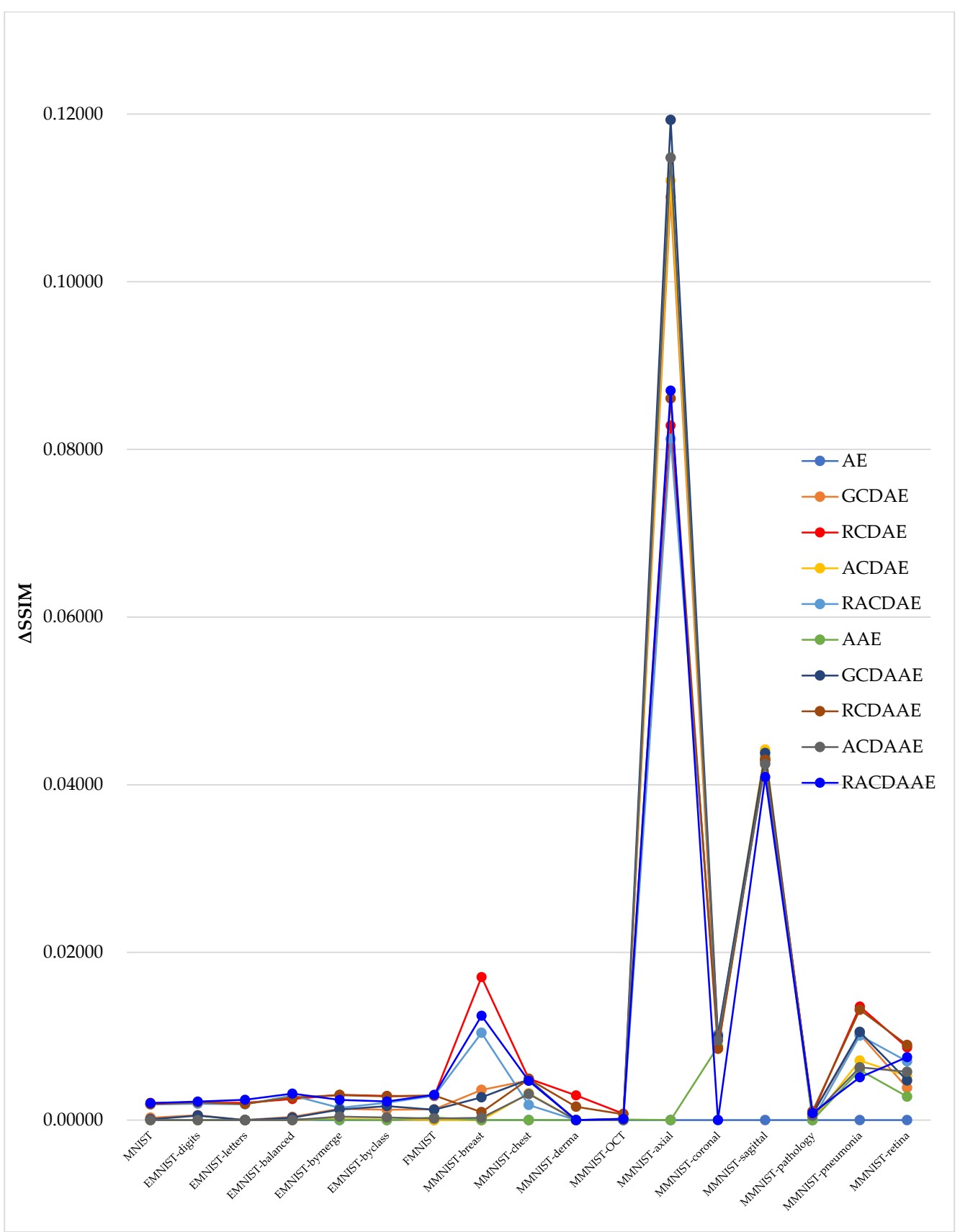

**Figure 15.** Experimental results of different algorithms on the four data sets.

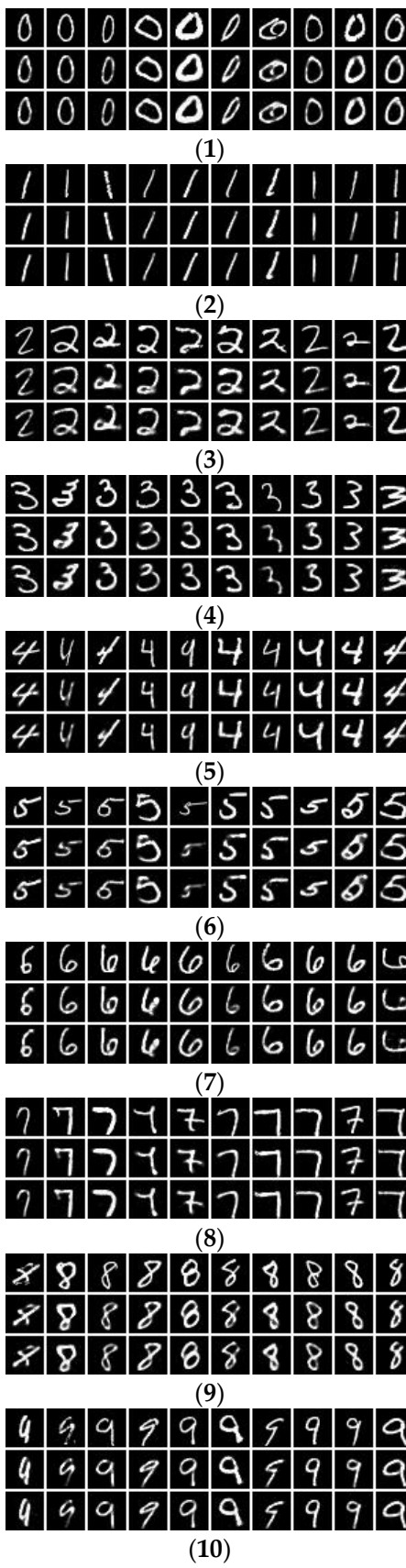

**Figure 16.** Reconstruction images on the MNIST data set. For each subfigure, the top row shows the original images, the middle row shows the recovery images of AE, and the bottom row shows the recovery images of RCDAE.

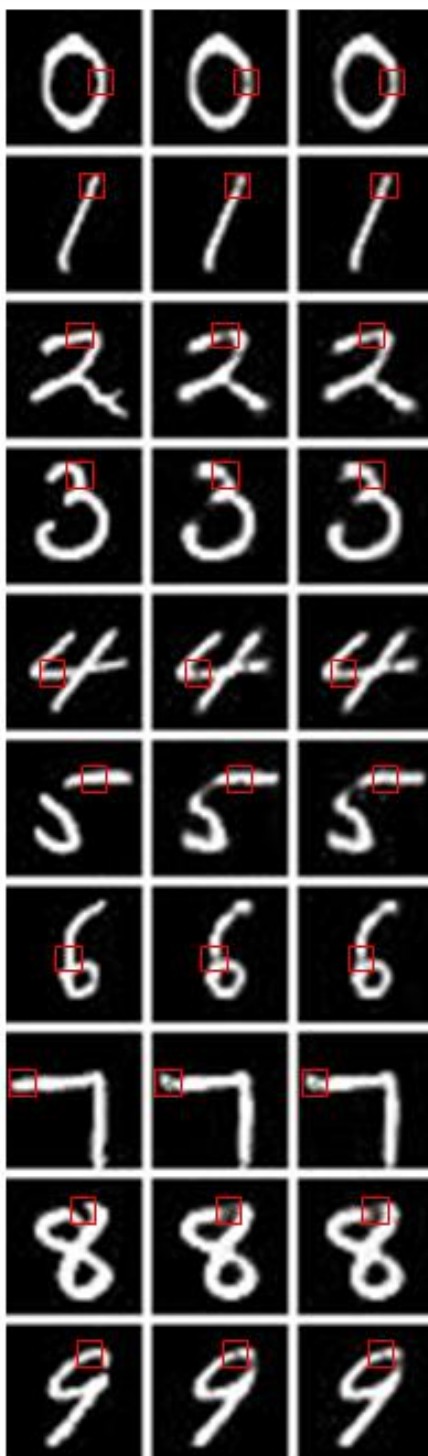

**Figure 17.** Marked reconstruction images on the MNIST data set.

The recovery images on the MNIST data set are shown in Figure 16. For each subfigure in Figure 16, the top row shows the original images, the middle row shows the recovery images of AE, and the bottom row shows the recovery images of RCDAE. It is not easy to find the SSIM differences between AE and RCDAE in Figure 16. Therefore, the marked re-establishment images on the MNIST data set are illustrated in Figure 17. The left column shows the original images, the middle column shows the re-establishment images of AE, and the right column shows the re-establishment images of RCDAE. It is easy to notice the SSIM differences between AE and RCDAE in the red marked squares in Figure 17.

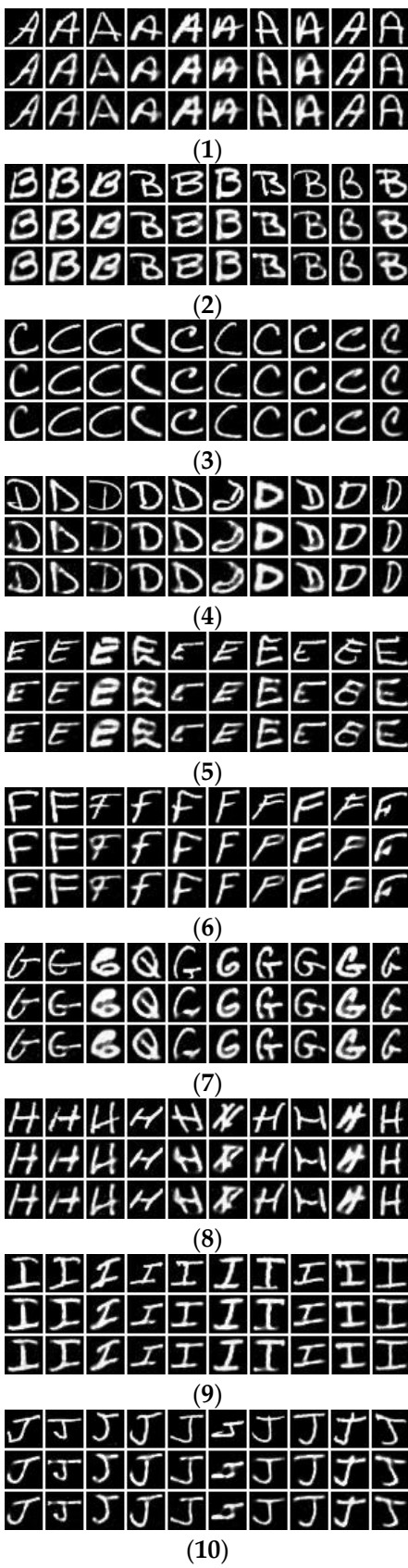

**Figure 18.** Reconstruction image on the EMNIST data set (**big letters**). For each subfigure, the top row displays the original images, the middle row displays the recovery images of AE, and the bottom row displays the recovery images of RCDAE.

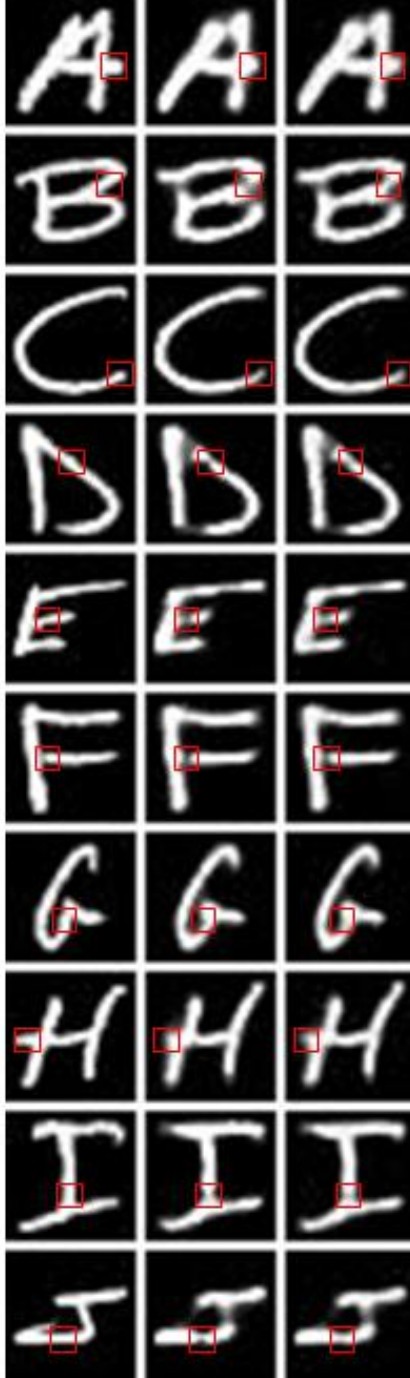

**Figure 19.** Marked reconstruction images on the EMNIST data set (**big letters**).

Similarly, the reconstitution images on the EMNIST data set (big letters) are demonstrated in Figure 18; the marked reconstitution images on the EMNIST data set (big letters) are demonstrated in Figure 19. The rebuilding images on the EMNIST data set (small letters) are displayed in Figure 20; the marked rebuilding images on the EMNIST data set (small letters) are displayed in Figure 21. The reconstruction images on the FMNIST data set are shown in Figure 22; the marked reconstruction images on the FMNIST data set are shown in Figure 23. The recovery images on the MMNIST data set are displayed in Figure 24; the marked recovery images on the MMNIST data set are displayed in Figure 25.

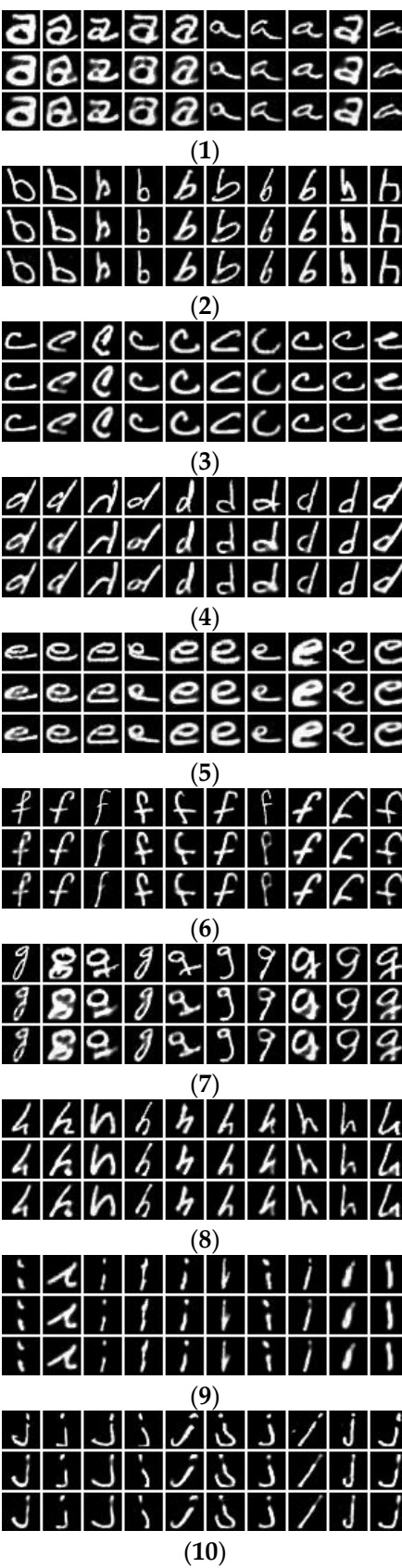

**Figure 20.** Reconstruction images on the EMNIST data set (**small letters**). For each subfigure, the top row exhibits the original images, the middle row exhibits the recovery images of AE, and the bottom row exhibits the recovery images of RCDAE.

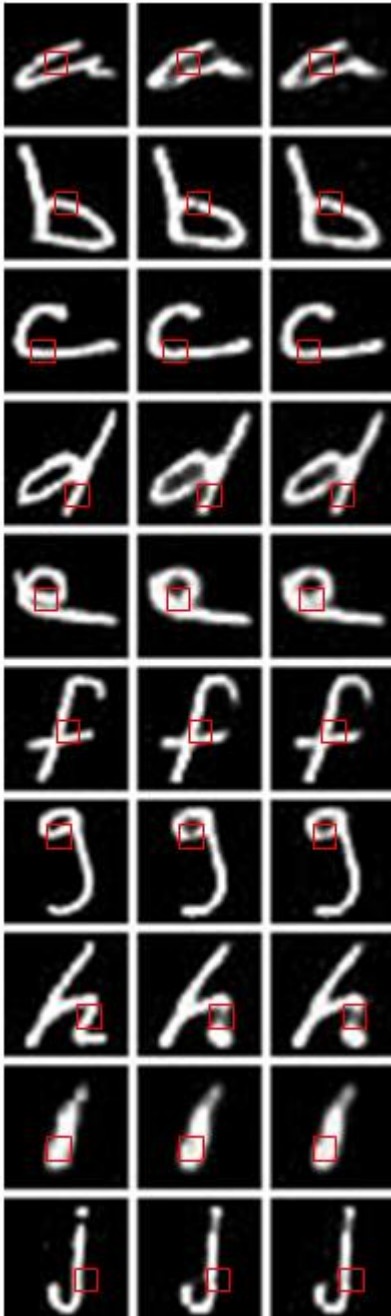

**Figure 21.** Marked reconstruction images on the EMNIST data set (**small letters**).

It is revealed from Figures 16–25 that the proposed algorithms achieve significant improvements in re-establishment performance on the MNST and EMNIST data sets. It is also manifested in Figures 16–25 that the proposed methods merely obtain unobvious promotion of re-establishment performance on the FMNIST and MMIST data sets. For instance, in the first row of Figure 25, the difference between the proposed and classical methods can only be found after enlarging the images; in the eighth row of Figure 25, conspicuous differences still cannot be found even after enlarging the images. Nevertheless, both of them are the true experimental results, which should be accepted and explained. The lack of differences in these results is attributed to four reasons. The first reason is that the quality of the original images is low on the FMNIST and MMNIST data sets. The second reason is that only the illumination component of original color images on part of the MMNIST data sets is reserved. The reconstruction performance will be improved

if the original color images are utilized. The third reason is that the dimension of latent space is 30. It a very low choice compared with 784 (28 × 28), the dimensions of the original image. The fourth reason is that the convolutional layer is not utilized in the architecture of auto-encoders. For the purpose of decreasing the computational load, the convolutional layer was not adopted in the proposed approaches. The convolutional layer can effectively extract image features and reconstruct the original image, and is expected to further improve the reconstruction performance of the proposed approaches; this will be investigated in our future research.

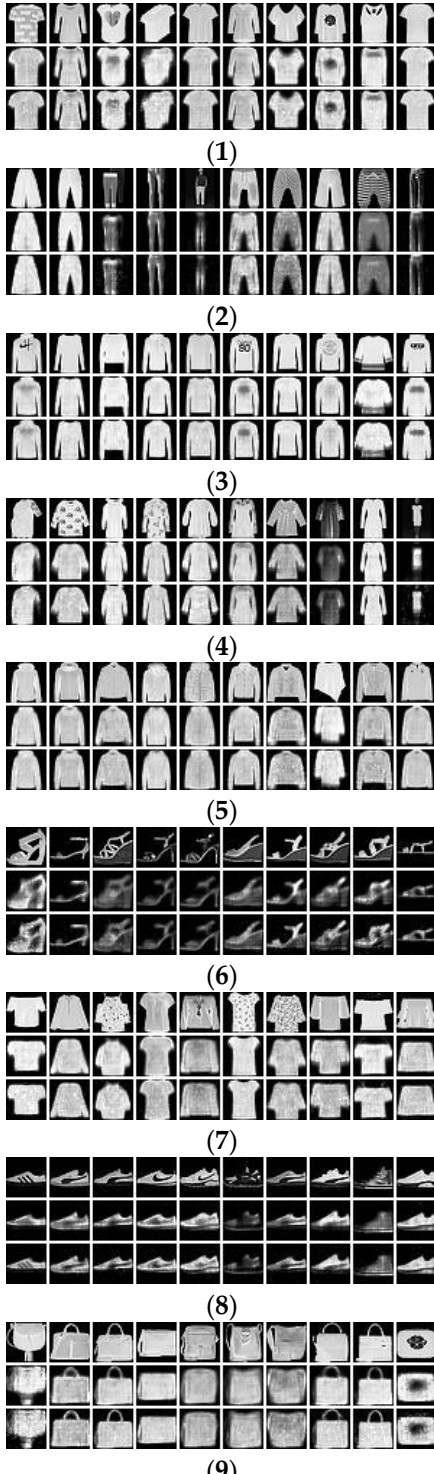

**Figure 22.** *Cont.*

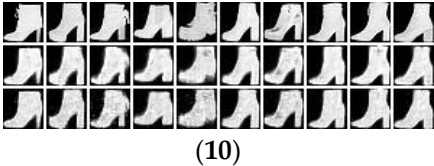

(**10**)

**Figure 22.** Reconstruction images on the FMNIST data set. For each subfigure, the top row demonstrates the original images, the middle row demonstrates the recovery images of AE, and the bottom row demonstrates the recovery images of RCDAE.

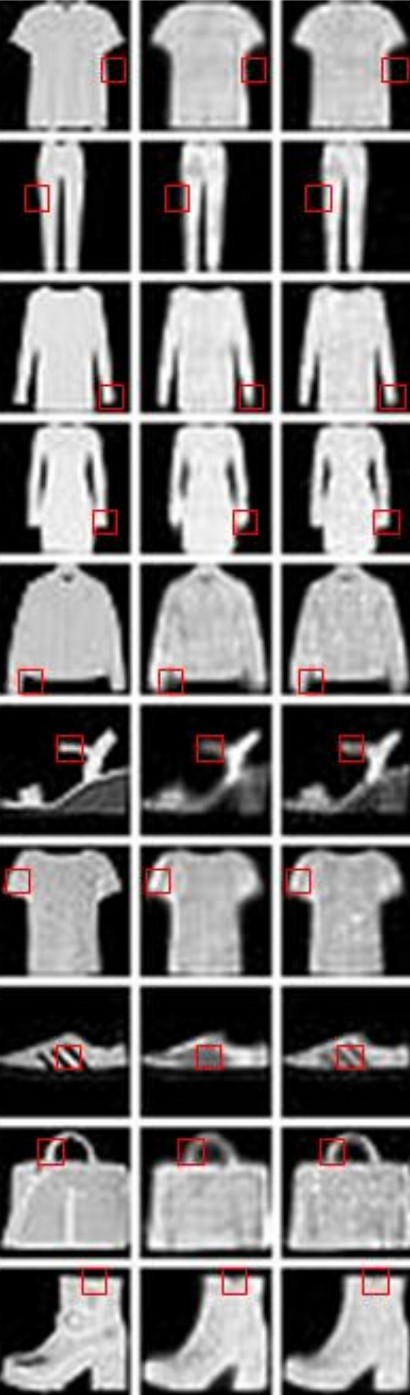

**Figure 23.** Marked reconstruction images on the FMNIST data set.

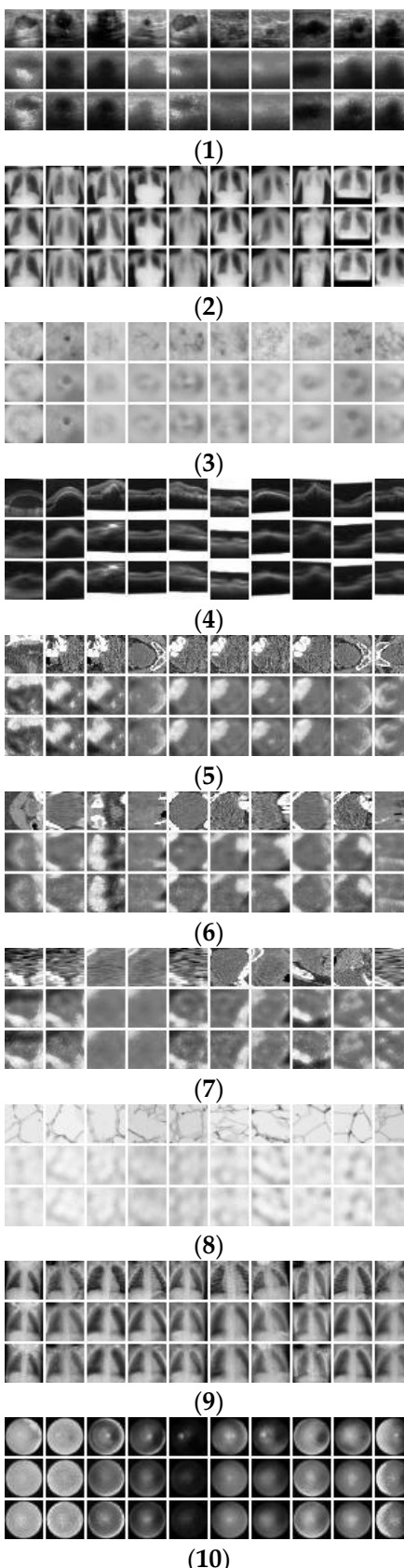

**Figure 24.** Reconstruction images on the MMNIST data set. For each subfigure, the top row reveals the original images, the middle row reveals the recovery images of AE, and the bottom row reveals the recovery images of RCDAE.

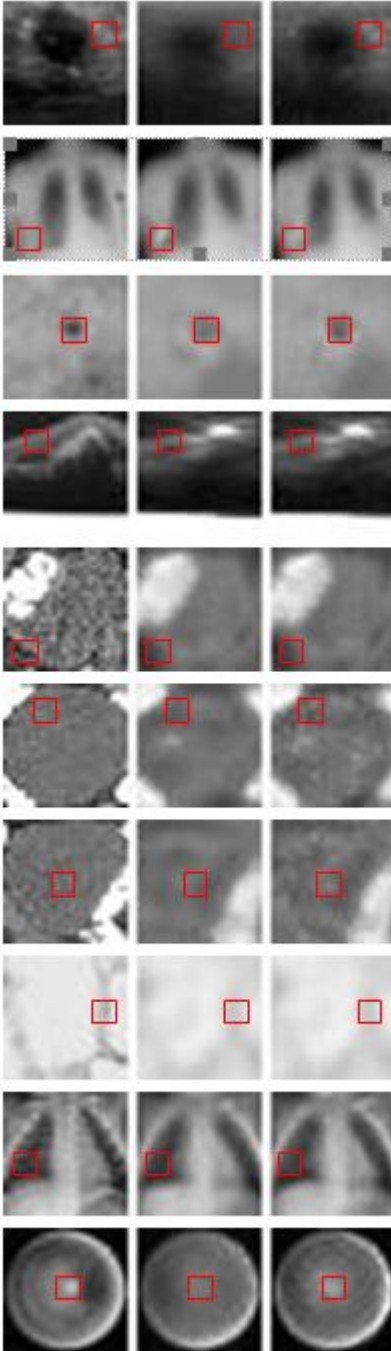

**Figure 25.** Marked reconstruction images on the MMNIST data set.

## 5. Conclusions

This paper proposes cascade decoders-based auto-encoders for image reconstruction. They comprise the architecture of multi-level decoders and related optimization problems and training algorithms. This article concentrates on the classical AE and AAE, as well as their serial decoders-based versions. Residual learning and adversarial learning are contained in the proposed approaches. The effectiveness of cascade decoders for image reconstruction is demonstrated in mathematics. It is evaluated based on the experimental results on four open data sets that the proposed cascade decoders-based auto-encoders are superior to classical auto-encoders in the performance of image reconstruction. In particular, residual learning is well suited for image reconstruction.

In our future research, experiments on data sets with large resolution images and colorful images will be conducted. Experiments on other advanced auto-encoders, such

as VAE and WAE, will also be explored. The convolutional layer or transformer layer will be introduced into the proposed algorithms. The constraints on high-dimensional reconstruction data, such as sparse and low-rank priors, will be utilized to advance the reconstruction performance of auto-encoders. Generalized auto-encoders-based data compression and signal-compressed sensing will also be probed. The auto-encoders-based lossless reconstruction will further be studied.

## 6. Patents

The patent with application number CN202110934815.7 and publication number CN113642709A results from the research reported in this manuscript.

**Author Contributions:** Conceptualization, H.L. and M.T.; methodology, H.L. and D.G.; writing, H.L.; supervision, M.S. All authors have read and agreed to the published version of the manuscript.

**Funding:** This research received no external funding.

**Institutional Review Board Statement:** Not applicable.

**Informed Consent Statement:** Not applicable.

**Data Availability Statement:** Not applicable.

**Acknowledgments:** The authors would very much like to thank Yui Chun Leung for the collection of MATLAB implementations of Generative Adversarial Networks (GANs) on the GitHub website (https://github.com/zcemycl/Matlab-GAN, accessed on 26 November 2020). We took advantage of the AAE codes and halved the dimension of AAE latent space.

**Conflicts of Interest:** The authors declare no conflict of interest.

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
