# Peer review of "Serial Decoders-Based Auto-Encoders for Image Reconstruction"

_applsci, doi:10.3390/app12168256_

Round 1

Reviewer 1 Report

Thanks for the submitted paper. The paper proposes a serial decoder-based autoencoder for lossless image reconstruction. 

Major points:

1. The major concern is that the quality of the paper needs to be improved.    Please revise the paper with a professional English editing service.

2. The main challenge of the current machine learning framework is how to address the biased data. Please discuss the potential implication of the proposed method on biased data. 

3. Elaborate on how progressive training could improve variability and stability. The improved stability may interest readers. Consider highlighting the advantage in the abstract and introduction. 

Minor points:

1. The English style is unusual. The language must be edited by a professional editing service. Otherwise, the message of the paper cannot be efficiently delivered.

2. Fig. 1 is unclear. Please label the x-axis.

3. The introduction is too long. Consider outlining the autoencoders in a table.

4. Fig 25-26: It is strange to demonstrate low-resolution medical images in MMNIST. To avoid possible misunderstanding, please consider using high-resolution images or removing medical images.

5. The MATLAB version and computer equipment are not specified.

6. Figures 13 - 15: Texts in the figure are 90 degrees rotated to the left.

Author Response

Point 1: Thanks for the submitted paper. The paper proposes a serial decoder-based autoencoder for lossless image reconstruction.

Response 1: We express our deep thanks to the reviewer for the laborious review work on this paper which focuses on loss image reconstruction and provides the potential basis of lossless image reconstruction.

Major points:

Point 2: 1. The major concern is that the quality of the paper needs to be improved.  Please revise the paper with a professional English editing service.

Response 2: As a non-native writer, the corresponding author tries his best to polish the paper in order to avoid possible typographical and grammatical errors to a great degree. In addition, it is inconvenient for us to find a suitable editing service during the short revision period of 10 days.

Point 3: 2. The main challenge of the current machine learning framework is how to address the biased data. Please discuss the potential implication of the proposed method on biased data.

Response 3: Biased data have smaller dynamic range compared with original image data. It is more effortless for a deep neural network to learn the biased data compared with the primary data. Hence, the proposed learning framework depending on biased data holds better reconstruction performance.

Point 4: 3. Elaborate on how progressive training could improve variability and stability. The improved stability may interest readers. Consider highlighting the advantage in the abstract and introduction.

Response 4: The proposed training method gradually increases the decoders of autoencoders. It is hard for us to train stable autoencoders with multiple decoders and large hyper-parameters. However, It is easy for us to train a stable unit of autoencoders with single decoder and small hyper-parameters. A decoder can merely learn low image variation, but serial decodes can learn high image variation. Hence, progressively training can efficiently strengthen the quality, stability and variability of image reconstruction. The abstract and introduction sections are updated according to the comment.

Minor points:

Point 5: 1. The English style is unusual. The language must be edited by a professional editing service. Otherwise, the message of the paper cannot be efficiently delivered.

Response 5: We do our best to promote the writing quality of this paper. The whole paper is carefully checked to avert potential typographical and mathematical mistakes.

Point 6: 2. Fig. 1 is unclear. Please label the x-axis.

Response 6: The x-axis label has already existed at the bottom of Fig. 1 in the previous version of this paper.

Point 7: 3. The introduction is too long. Consider outlining the autoencoders in a table.

Response 7: New tab. 1 is supplemented into the introduction section for description clearness.

Point 8: 4. Fig 25-26: It is strange to demonstrate low-resolution medical images in MMNIST. To avoid possible misunderstanding, please consider using high-resolution images or removing medical images.

Response 8: Because the original resolution of MMNIST dataset is very low and Fig. 25-26 are the true experimental results, we would like to preserve the actual images in Fig. 25-26.

Point 9: 5. The MATLAB version and computer equipment are not specified.

Response 9: The experimental software platform is MATLAB 2020b on Windows 10 or Linux. For the small datasets, MNIST, FMNIST and MMNIST, the experimental hardware platform is a laptop with 2.6 GHz dual-core processor and 8GB memory; For the big dataset, EMNIST, the experimental hardware platform is a super computer with high-speed GPUs and huge memory.

Point 10: 6. Figures 13 - 15: Texts in the figure are 90 degrees rotated to the left.

Response 10: The text directions in Fig. 13-15 are adjusted according to the comment.

Reviewer 2 Report

1- Most of the article is similar to the previous article of the authors. It appears as a warning in plagiarism programs. The similarity value of 65% is quite high. For this reason, similar parts need to be reduced or rewritten.

2- The findings mentioned in the content are very small and difficult to select. It would be nice if it was presented with higher resolution or larger images. It would be good to see the code from which the findings were obtained.

3- Benchmarking and code content should be shared with reviewers for verification and review of the proposed method. Related setup, libraries and versions should also be shared. For this purpose, a colab can also be opened or a sample video can be presented.

Author Response

Point 1: 1- Most of the article is similar to the previous article of the authors. It appears as a warning in plagiarism programs. The similarity value of 65% is quite high. For this reason, similar parts need to be reduced or rewritten.

Response 1: The high repetition is due to that fact that this paper has been submitted to www.arxiv.com (https://doi.org/10.48550/arXiv.2107.00002). However, it is only a preprint version, not a formal publishing.

Point 2: 2- The findings mentioned in the content are very small and difficult to select. It would be nice if it was presented with higher resolution or larger images. It would be good to see the code from which the findings were obtained.

Response 2: Although the original datasets employed in this paper hold small images with low resolution, they are widely-used well-known open datasets and efficiently demonstrate the performance of the proposed methods. Datasets with large and high-resolution images will be considered in our future work. In the current of this paper, the difference between original and reconstructed images are found by hand. We plan to write and publish automatic codes of finding image difference in our future work.

Point 3: 3- Benchmarking and code content should be shared with reviewers for verification and review of the proposed method. Related setup, libraries and versions should also be shared. For this purpose, a colab can also be opened or a sample video can be presented.

Response 3: This paper is based on open-source codes (https://github.com/zcemycl/Matlab-GAN). We take advantage of the AAE codes and halve the dimension of AAE latent space. The updated codes of this paper is released on GitHub: https://github.com/hongguili/Serial-Decoders-Based-Autoencoders.

Reviewer 3 Report

The article is well-written, good in quality, and in the perspective of novelty, accepted as it is for publication

Author Response

Point 1: The article is well-written, good in quality, and in the perspective of novelty, accepted as it is for publication.

Response 1: We thank very much the reviewer for the positive feedback.

Round 2

Reviewer 1 Report

Thank you for revisiting the manuscript.

However, a review is not your language advisor, nor is a rubber stamp. Since the paper has multiple grammatical errors, e.g. inappropriate use of verb., sentences separated by commas. Write sentences in paragraphs. 

Reviewer 2 Report

Almost everything in the presented text has already been published. Few innovations were submitted for publication in the journal. The decision on this matter belongs to the editor. The research is methodologically appropriate. The desire to enlarge the paper images and to mark them more legible has been realized, (I do not mean to increase the resolution of the existing database here.). Performance comparison with alternative and competing methods is a must, and the authors have provided this. I think the publication is acceptable in its current state.